# Hierarchical Fine-grained Preference Optimization for Physically Plausible Video Generation

**Harold Haodong Chen**[1,2,3], **Haojian Huang**[1,2],
**Qifeng Chen**[2,3], **Harry Yang**[†2,3], **Ser-Nam Lim**[†3,4]
[1]The Hong Kong University of Science and Technology (Guangzhou)
[2]The Hong Kong University of Science and Technology
[3]Everlyn AI, [4]University of Central Florida
[†]Corresponding Author
✉ haroldchen328@gmail.com

## Abstract

Recent advancements in video generation have enabled the creation of high-quality, visually compelling videos. However, generating videos that adhere to the laws of physics remains a critical challenge for applications requiring realism and accuracy. In this work, we propose **PhysHPO**, a novel framework for Hierarchical Cross-Modal Direct Preference Optimization, to tackle this challenge by enabling fine-grained preference alignment for physically plausible video generation. PhysHPO optimizes video alignment across four hierarchical granularities: a) ***Instance Level***, aligning the overall video content with the input prompt; b) ***State Level***, ensuring temporal consistency using boundary frames as anchors; c) ***Motion Level***, modeling motion trajectories for realistic dynamics; and d) ***Semantic Level***, maintaining logical consistency between narrative and visuals. Recognizing that real-world videos are the best reflections of physical phenomena, we further introduce an automated data selection pipeline to efficiently identify and utilize *"good data"* from existing large-scale text-video datasets, thereby eliminating the need for costly and time-intensive dataset construction. Extensive experiments on both physics-focused and general capability benchmarks demonstrate that PhysHPO significantly improves physical plausibility and overall video generation quality of advanced models. To the best of our knowledge, this is the first work to explore fine-grained preference alignment and data selection for video generation, paving the way for more realistic and human-preferred video generation paradigms. PhysHPO Page

## 1 Introduction

Video generation has recently achieved significant strides in producing high-quality, visually compelling [39, 73, 90, 99, 64], and lengthy [28, 19, 60, 96] videos depicting real-world scenarios. Despite these advancements, generating videos that adhere to the laws of physics remains a challenging and critical research problem. The ability to create physically plausible videos is essential for applications ranging from virtual reality to simulations, where realism and accuracy are paramount.

Current efforts to enhance the physical fidelity of text-to-video (T2V) generation can be broadly categorized into **test-time reflection-based optimization** and **training-time tuning-based optimization**. Test-time reflection-based methods, such as PhyT2V [85], employ a large language model (LLM) to iteratively refine initial T2V prompts. While effective, these methods significantly decrease computational efficiency and are inherently limited by the upper bounds of the model's capabilities in self-correction paradigms. Conversely, recent tuning-based methods (*e.g.*, WISA [75] and SynVideo [98]) focus on traditional supervised fine-tuning (SFT) paradigms. Although effective, SFT relies

39th Conference on Neural Information Processing Systems (NeurIPS 2025).

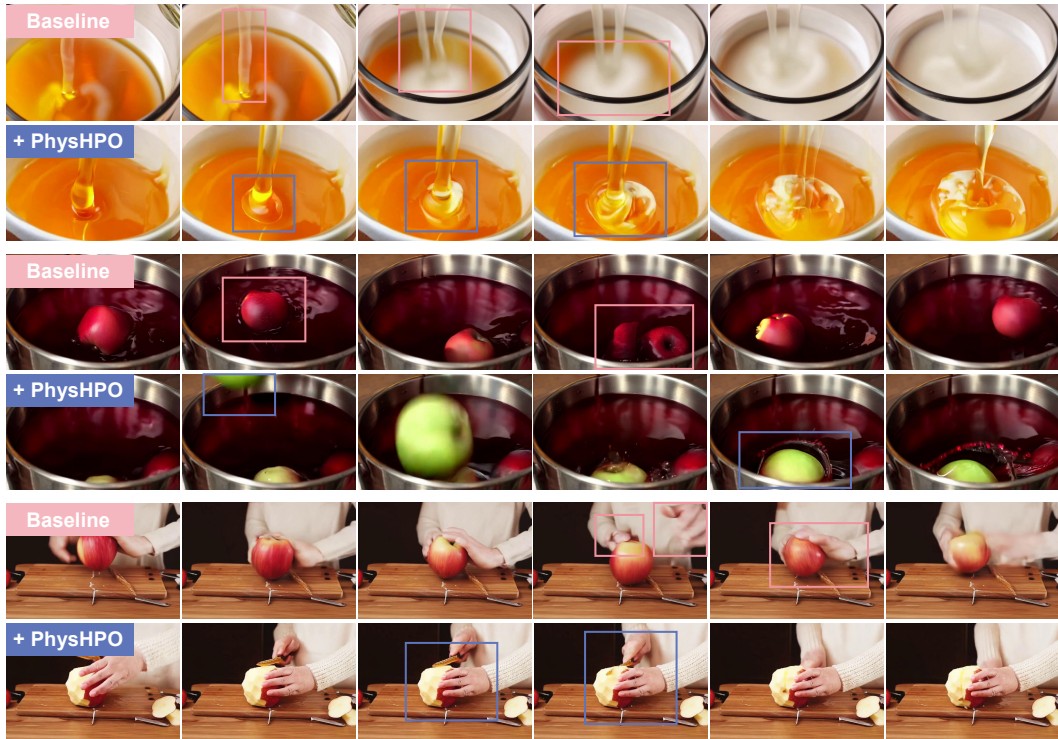

Figure 1: PhysHPO significantly improves the physical plausibility of video generation. Text prompts are adopted from VideoPhy [6]. (***Top***) **Fluid-Fluid**: *Honey diffusing into warm milk.* (***Middle***) **Solid-Fluid**: *An apple falls into a vat of red wine.* (***Bottom***) **Solid-Solid**: *Peeler peels an apple.*

heavily on fixed supervisory signals, which show suboptimal effectiveness when targeting specific capability optimization [43, 51, 23].

Emerging post-training techniques like Direct Preference Optimization (DPO) [61] have demonstrated more efficient pair-wise optimization paradigms to discern differences between human-preferred and non-preferred samples, enhancing specific aspects of visual generation models, *e.g.*, safety [63, 50], customization [43], super-resolution [9]. This indicates DPO holds potential for physically plausible video generation, yet remains largely underexplored. However, recent DPO works in video generation [51, 94, 36, 16] predominantly focus on coarse-grained alignment between videos at the instance level, which may result in suboptimal preference alignment [31]. We emphasize that achieving optimal alignment, particularly for physically plausible video generation, necessitates fine-grained preference alignment that goes beyond visual appeal to incorporate detailed modeling.

Building on these insights, we propose a novel framework, *Hierarchical Cross-Modal Direct Preference Optimization* for physically plausible video generation, namely **PhysHPO**. PhysHPO enhances video preference alignment across hierarchical granularities. Specifically, we design four levels of alignment: ❶ *Instance Level*: Ensuring comprehensive alignment by matching the overall prompt content with the most suitable video. ❷ *State Level*: Leveraging boundary frames as critical anchors for establishing plausible states. ❸ *Motion Level*: Modeling motion through structural information within videos, enabling alignment beyond mere pixel appearance. ❹ *Semantic Level*: Ensuring logical consistency between what is described and what is visually portrayed.

Moreover, compared to existing SFT methods [75, 98] for generating physically plausible videos, which consume extensive resources in dataset construction, we argue that the popular *"One-Model-One-Dataset"* paradigm may not be optimal. Unlike tasks like condition-guided (*e.g.*, dancing [103, 30]) or style-focused (*e.g.*, cartoon [82, 19]) video generation, which demand additional data annotations or domain-specific data, real-world videos inherently encapsulate rich physical dynamics, suggesting potential for more efficient data utilization. To this end, we propose a novel automated data selection pipeline to efficiently process existing large-scale text-video datasets, circumventing exhaustive new data collection efforts for physically plausible video generation. Unlike existing data processing pipelines for high-quality video large-scale pre-training (*e.g.*, in Open-Sora [99]), our key idea is to select a subset of "good data" from large, high-quality raw data that closely matches desired

target requirements—intuitively, real-world videos where physical laws are prominently reflected. To the best of our knowledge, no prior work has explored data selection in the domain of video generation. To summarize, this work contributes in threefold:

- We introduce a novel Hierarchical Cross-Modal DPO (**PhysHPO**) framework for video generation, a more fine-grained DPO strategy to enhance alignment between videos, optimizing for physically plausible video generation.

- We advocate leveraging real-world videos rather than constructing datasets from scratch for physically plausible video generation. This approach intuitively reflects physical phenomena and introduces the data selection problem to video generation for the first time.

- Extensive experiments on both physics-focused (*i.e.*, VideoPhy [6], PhyGenBench [54]) and general capability (*i.e.*, VBench [34]) benchmarks demonstrate that PhysHPO significantly improves the physical plausibility and overall video generation capabilities of existing advanced models.

## 2 Related Work

**Physics-aware Video Generation**   While generating visually compelling videos has advanced, achieving physics plausibility remains challenging, as noted by users and benchmarks [6, 38, 56, 42, 54, 65]. Existing works [40, 55, 52, 88] in image-to-video (I2V) focus on parsing objects from images and estimating their motion by considering physical properties, and Li et al. [41] explores solely object freefall. However, these methods are limited to fixed physical categories or static scenarios, which restricts their generalizability. Recent works [85, 75, 98] aim to enhance the broader physical plausibility of T2V models. PhyT2V [85] introduces an LLM for iteratively prompt refinement during test time. However, the hugely increased inference overhead and inherent performance limitations restrict its effectiveness. Subsequent research has explored traditional SFT to improve model performance. Specifically, WISA [75] constructs a 32K video dataset, and SynVideo [98] uses a computer graphics pipeline to synthesize video data, both requiring substantial manual intervention, which is resource-intensive. Intuitively, real-world videos naturally reflect physical phenomena. Thus, efficiently utilizing existing datasets without unnecessary data inefficiency is an intriguing problem. To this end, we introduce the concept of data selection for video generation for the first time, automating the process to leverage available data resources effectively. While SFT has demonstrated superior performance in pre-training, we propose adopting the DPO post-training paradigm to further model differences between video pairs, enabling deeper exploration of physical information.

**Data Selection for "Good Data"**   Data selection is a pivotal technique for efficiently training models without sacrificing performance [3, 74], as highlighted in both pre-training [7, 70] and post-training [14, 47] stages. Recent studies emphasize that the effectiveness of LMs stems from a combination of large-scale pre-training and smaller, meticulously curated instruction datasets [69, 17, 18, 77, 101]. Various sophisticated approaches have been proposed for LMs, including quality-based [45, 20, 46], diversity-aware [24, 53], complexity considerations [84, 67, 35], alongside simpler heuristics like selecting longer responses and gradient-based coreset selection [80, 95, 59], collectively demonstrating significant benefits in LM training. However, existing methods predominantly target LMs, leaving their potential in video generation largely unexplored. In this work, we pioneer the exploration of data selection strategies specifically for video generation in the post-training phase. We propose a new automated data selection pipeline, explicitly emphasizing *reality*, *physical fidelity*, and *diversity* to enhance the generation of physically plausible videos.

**Direct Preference Optimization for Video Generation**   DPO [61] has emerged as a promising alternative to traditional RLHF [58] for enhancing LMs [31, 27, 49] and image generative models [72, 78] without needing an extra reward model. Recent efforts have applied DPO to video generation. Pioneering work like VideoDPO [51] follows the Diffusion-DPO [72] paradigm, introducing OmniScore for adaptive video scoring, while HuViDPO [36] aligns outputs with human preferences using feedback. OnlineVPO [94] optimizes off-policy with a video-centric model, and MagicID [43] uses DPO for ID-conditioned customization. Cheng et al. [16] employs a discriminator-free approach for direct optimization, and GAPO [102] introduces AnimeReward for anime video generation. Recent foundation models like Seaweed-7B [64], SkyReels-V2 [10] also incorporate DPO in post-training, further highlighting its potential. However, existing DPO works focus solely on coarse-grained alignment at the video instance level, overlooking finer details. In this paper, we aim to enhance DPO's effectiveness, specifically for generating physically plausible videos by modeling fine-grained alignment. To achieve this, we propose a novel *hierarchical cross-modal preference optimization* framework, named PhysHPO, which effectively captures the fine-grained details of videos.

## 3 Preliminaries

**Direct Preference Optimization for Diffusion Models** Diffusion-DPO [72] adapts human preference alignment to iterative generation processes, eliminating explicit reward modeling. Given a diffusion model $p_\theta(y|x,t)$ and reference model $p_{\text{ref}}(y|x,t)$, the denoising objective with preference constraints becomes

$$\max_{p_\theta} \mathbb{E}_{t,x,y\sim p_\theta}[r(x,y,t)] - \beta D_{\text{KL}}\left(p_\theta(y|x,t) \parallel p_{\text{ref}}(y|x,t)\right), \quad (1)$$

where $r(\cdot)$ denotes the time-dependent reward function, $x$ the input condition, $y$ the generated sample, and $t$ the timestep. DPO establishes a trajectory-level reward mapping through denoising paths:

$$r(x,y,t) = \beta \log \frac{p_\theta(y|x,t)}{p_{\text{ref}}(y|x,t)} + \beta \log Z(x,t), \quad (2)$$

where $\beta$ controls KL constraint strength, and $Z(x,t)$ the time-dependent partition function.

The preference optimization objective is derived by substituting the reward parameterization and applying negative log-likelihood loss:

$$\mathcal{L}_{\text{DPO}} = -\mathbb{E}_{(x,y_w,y_l)\sim\mathcal{D}} \log \sigma\left(u(x,y_w,y_l,t)\right), \ u = \underbrace{\log \frac{p_\theta(y_w|x,t)}{p_{\text{ref}}(y_w|x,t)}}_{\text{Preferred path score}} - \underbrace{\log \frac{p_\theta(y_l|x,t)}{p_{\text{ref}}(y_l|x,t)}}_{\text{Non-preferred path score}}. \quad (3)$$

## 4 Data Selection: From Physical to Diverse

Existing text-video datasets have already been rigorously screened for quantity and quality [99, 57, 48, 15]. While creating specific datasets for different tasks is popular [93, 33, 19, 92], for physically plausible video generation, real-world videos inherently reflect physical laws. Thus, selecting existing data may be more optimal than constructing new datasets [75, 98]. In this section, we present an initial exploration of the data selection in video generation, focusing specifically on identifying "good data" that effectively reflects physical laws.

### 4.1 The Data Selection Problem

Video generation models undergo pre-training to learn world knowledge [8, 2]. Fine-tuning these models aligns them with specific goals, analogous to instruction tuning in LMs [91]. Data selection has proven effective for aligning LMs by identifying a small, high-quality dataset [53, 81, 25, 80]. Consider a large data pool $D = \{x_1, x_2, \cdots, x_n\}$, where each data point $x_i = (V_i, C_i)$ consists of a video $V_i$ and its corresponding caption $C_i$. The objective is to select a subset $S^{(m)}$ of size $m$ that maximizes the post-training performance $\mathcal{P}$:

$$S^* = \arg\max \mathcal{P}(S^{(m)}). \quad (4)$$

While there are various potential ways for data selection in a desired domain, our goal is to keep the process *as simple as possible to be practical*, as shown in Figure 2 (*Left*). We next detail the process.

### 4.2 Selection for Real-World Physics

**Reality** Given a large-scale high-quality text-video dataset $D$ (we adopt OpenVidHD-0.4M [57] in this work, which is widely utilized for post-training [12, 87, 68, 13]), the first step is to filter out real-world videos into a real-world data pool $D'$. Specifically, considering the significant content differences between real-world and virtual videos, we employ a vision-

Table 1: Accuracy of two-stage reality selection with 1,000 samples.

| Reality Selection | Acc (%) |
|---|---|
| + Qwen2.5-VL [4] | 99.6 |
| + DeepSeek-VL2 [79] | 100.0 |

language models (VLMs) (*i.e.*, Qwen2.5-VL [4]) to determine whether a video $V_i$ is a real-world video. We further task DeepSeek-VL2 [79] for a double-check to ensure the accuracy. Table 1 shows the accuracy from a random sample of 1,000 videos from $D$.

**Physical Fidelity** To align with previous works, we adopt 17 physical phenomena across three physics categories from WISA [75]: ❶ **Dynamic** (rigid body motion, collision, liquid motion, gas motion, elastic motion, deformation), ❷ **Thermodynamic** (melting, solidification, vaporization, liquefaction, combustion, explosion), and ❸ **Optic** (reflection, refraction, scattering, interference and diffraction, unnatural light source). Unlike widely used heuristic classification-based data selection methods in LMs that require a target dataset for reference [81, 89, 21], we follow the popular LLM-as-a-Judge paradigm [53, 26, 102] to employ LLMs for automatic data evaluation.

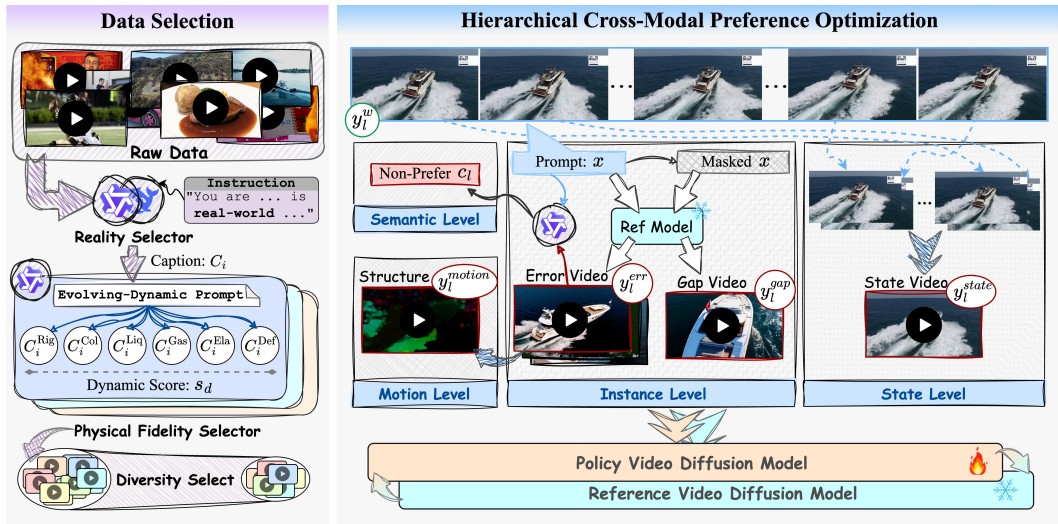

Figure 2: The overview of our proposed (**Left**) data selection and (**Right**) PhysHPO framework.

To minimize consumption, we then perform selections at the caption level. Given a video caption $C_i$, a naive way is to task an LLM with evaluating whether the sample clearly reflects physical laws by directly assigning a score. However, LLMs might assign similar scores to most samples due to the lack of references [32]. To address this, we propose adopting *in-depth evolving prompting* [84] to augment captions. Unlike previous works [53, 84] that augment samples in a single dimension (*e.g.*, complexity), we explore multiple dimensions. Specifically, as illustrated in Figure 2, we first use the crafted *evolving-dynamic* prompts with an LLM (*i.e.*, Qwen2.5 [86]) to augment the caption, emphasizing each of the seven physical phenomena in the "Dynamic" category. This helps the model distinguish differences in captions more precisely, achieving fine-grained scoring. We then ask the LLM to rank and score these seven samples, obtaining the "Dynamic" scores $s_d$ corresponding to the caption $C_i$. Similarly, the "Thermodynamic" score $s_t$ and "Optic" score $s_o$ can be obtained. This strategy provides a more nuanced distinction of physical phenomena. Details are in **Appendix §B**.

The total score is calculated as $s = s_d \times s_t \times s_o$. Then all samples in $D'$ are sorted based on $s$, resulting in the sorted pool $S' = \{x'_1, x'_2, \cdots, x'_k\}$, where $x'_0$ is the sample with the highest score.

### 4.3 Selection for Diversity

To ensure an advanced generation model can handle varied user prompts, it's desirable for data to maintain maximum diversity within a given budget $m$. However, real-world data often exhibits redundancy [1]. To this end, we introduce an iterative method to ensure diversity in selected real-world data following Deita [53]. Briefly put, the key idea is to iteratively select samples $x'_i$ from the data pool $S'$, adding them to the dataset $S$ only if they contribute to its diversity. Formally, we define an indicator function $\mathbb{1}[\mathcal{F}(x'_i, S')]$, which is 1 if the diversity criterion $\mathcal{F}(x'_i, S')$ is met, otherwise 0. Using LLaMA-1 13B [71], captions are encoded into embeddings, and cosine distance $d$ is calculated between each candidate sample and its nearest neighbor in $S$. A sample contributes to diversity if $d < \tau$, with $\tau = 0.9$.

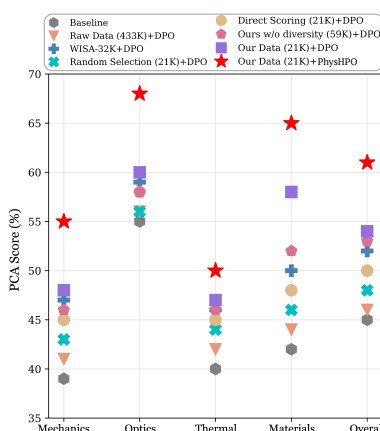

Figure 3: Performance comparison of different data strategies with our PhysHPO on PhysGenBench [54].

We compare the performance of different data selection strategies with vanilla DPO [72], along with our PhysHPO, as demonstrated in Figure 3. Several key observations are summarized: **Obs.❶** *Superiority of Data Selection.* Our data (21K), selected for reality and physical fidelity, demonstrates superior quality compared to other strategies like direct score, random selection, or even the manually constructed dataset WISA-32K [75] and unselected raw data (433K). "`Our Data (21K)+DPO`" outperforms "`Direct Scoring (21K)+DPO`", "`Random Selection (21K)+DPO`", "`WISA-32K+DPO`", and "`Raw Data (433K)+DPO`", highlighting the effectiveness of our selection. **Obs.❷** *Importance of Data Diversity.* The comparison between "`Our

Data (21K)+DPO" and "Ours w/o diversity (59K)+DPO" underscores the critical role of diversity in data selection. Despite fewer samples, the diverse dataset outperforms the larger one, indicating that diversity enhances model capability. **Obs.❸** *Strategic Data and Optimization Synergy.* The integration of our selected data with PhysHPO ("Our Data (21K)+PhysHPO") achieves the highest performance, illustrating the powerful synergy between strategic data selection and advanced optimization. More analysis is provided in **Appendix** §B. We next detail our PhysHPO framework.

## 5 Methodology

To tackle the challenging physically plausible video generation, we propose PhysHPO, which implements hierarchical preference optimization across four levels (Figure 2): (*i*) Instance-level Overall Preference Optimization, aligning video-level preferences to ensure overall quality (§5.1); (*ii*) State-level Boundary Preference Optimization, anchoring physical states at the start and end frames for stability (§5.2); (*iii*) Motion-level Dynamic Preference Optimization, utilizing structural information to accurately model and align motion representation (§5.3); (*iv*) Semantic-level Consistency Preference Optimization, ensuring fine-grained coherence between the narrative and visuals (§5.4).

### 5.1 Instance-level Overall Preference Optimization

Unlike VideoDPO [51], which relies solely on its base model for preference pair video data $(y_w, y_l)$, limiting optimization to the model's capabilities, our approach leverages data selected through our data selection process as the preferred video $y_w$ and its text prompt $x$. For the non-preferred video $y_l$, recent works [102, 94] often generate these using the base model based on $x$, adhering to Eq.(3) as the objective function. However, complex dependencies in non-preferred videos are often overlooked. Typically, video generation faces two main issues: ① Imperfections or errors in physical aspects, *e.g.*, motion; ② Failure to fully express the prompt's information, *e.g.*, missing objects. To mitigate this, we introduce two types of non-preferred videos into the optimization process:

$$\mathcal{L}_{\text{Instance}} = -\mathbb{E}_{(x,y_w,y_l)\sim\mathcal{D}} \log \sigma \left( \beta \log \frac{p_\theta(y_w|x,t)}{p_{\text{ref}}(y_w|x,t)} - \beta \log \frac{p_\theta(y_l|x,t)}{p_{\text{ref}}(y_l|x,t)} \right), \tag{5}$$

where

$$\log \frac{p_\theta(y_l|x,t)}{p_{\text{ref}}(y_l|x,t)} \leftarrow \sum_{i\in\{err,gap\}} \beta_i \log \frac{p_\theta(y_l^i|x,t)}{p_{\text{ref}}(y_l^i|x,t)}. \tag{6}$$

Here, $y_l^{err}$ denotes the error-prone or imperfect non-preferred video that aligns semantically with the preferred sample. In contrast, $y_l^{gap}$ represents non-preferred videos with semantic differences from the preferred sample. Specifically, for constructing $y_l^{err}$, we generate three videos per $x$ using the base model and select the one most visually similar by similarity calculation to the preferred video $y_w$ as the rejected video $y_l^{err}$. For $y_l^{gap}$, we introduce random masking in prompt $x$ during generation to create semantic differences from $y_w$.

### 5.2 State-level Boundary Preference Optimization

While the instance-level alignment captures overall content, fine-grained alignment is essential for generating physically plausible videos, which previous approaches have often overlooked. In video generation, maintaining a consistent physical state from the beginning to the end of a sequence is essential for ensuring realism and coherence. The initial and final frames are particularly critical as they anchor the video's physical narrative. To this end, we propose the state-level boundary alignment to enhance the model's focus on the physical states at the start and end of videos. Specifically, we replace the first and last $N$ frames of the preferred video $y_w$ to construct the state-level non-preferred sample $y_l^{state}$. The state level objective function can be defined as

$$\mathcal{L}_{\text{State}} \sim \log \sigma \left( u_{\text{State}}(x, y_w, y_l^{state}, t) \right). \tag{7}$$

### 5.3 Motion-level Dynamic Preference Optimization

While the visual appearance of videos is crucial for quality, excessive focus on appearance may obscure the model's alignment with physical dynamics. To address this issue and enhance the model's learning of dynamic information such as motion, we propose the extraction of structural information (*e.g.*, optical flow) from both preferred $y_w$ and non-preferred $y_l$ (specifically $y_l^{err}$) videos. These structural features, commonly used in condition-guided video generation [83, 22, 100], allow for a more explicit representation of the physical motion differences between preferred and non-preferred samples. Following Eq.(7), the motion-level objective function $\mathcal{L}_{\text{Motion}}$ is derived

Table 2: Evaluation on physics-focused benchmarks, *i.e.*, VideoPhy [6], PhyGenBench [54]; and general quality benchmark, *i.e.*, VBench [34]. Vanilla DPO is implemented with our selected data. We **bold** the best results, and "↑" denotes that higher is better.

| Method | VideoPhy [6] | | | | PhyGenBench [54] | | | | | VBench [34] | | |
|---|---|---|---|---|---|---|---|---|---|---|---|---|
| | SS↑ | SF↑ | FF↑ | Over.↑ | Mechanics↑ | Optics↑ | Thermal↑ | Materials↑ | Overall↑ | Total↑ | Quality↑ | Semantic↑ |
| CogVideoX-2B [90] | 12.7 | 21.9 | 25.4 | 18.6 | 0.38 | 0.43 | 0.34 | 0.39 | 0.39 | 81.6 | 82.5 | 77.8 |
| + PhyT2V [85] | 14.1 | 19.9 | 28.6 | 18.9 | 0.45 | 0.48 | 0.34 | 0.50 | 0.45 | 82.0 | 82.9 | 78.4 |
| + Vanilla DPO [72] | 15.5 | 19.2 | 28.6 | 19.2 | 0.43 | 0.50 | 0.40 | 0.50 | 0.46 | 82.2 | 83.1 | 79.0 |
| + **PhysHPO (Ours)** | **20.4** | **24.7** | **42.9** | **25.9** | **0.50** | **0.56** | **0.47** | **0.58** | **0.53** | **82.5** | **83.3** | **79.3** |
| CogVideoX-5B [90] | 24.4 | 53.1 | 43.6 | 39.6 | 0.39 | 0.55 | 0.40 | 0.42 | 0.45 | 81.9 | 83.1 | 77.3 |
| + PhyT2V [85] | 25.4 | 48.6 | 55.4 | 40.1 | 0.45 | 0.55 | 0.43 | 0.53 | 0.50 | 82.3 | 83.3 | 78.3 |
| + Vanilla DPO [72] | 28.2 | 50.0 | 51.8 | 41.3 | 0.48 | 0.60 | 0.47 | 0.58 | 0.54 | 82.4 | 83.3 | 78.7 |
| + **PhysHPO (Ours)** | **32.4** | **54.1** | **58.9** | **45.9** | **0.55** | **0.68** | **0.50** | **0.65** | **0.61** | **82.8** | **83.7** | **79.3** |

from $u_{\text{Motion}}(x, y_w \to y_w^{motion}, y_l \to y_l^{motion}, t)$, where $y_w^{motion}$ and $y_l^{motion}$ represent structural information extracted from $y_w$ and $y_l$, respectively.

### 5.4 Semantic-level Consistency Preference Optimization

Most prior DPO works in diffusion models primarily rely on visual alignment for optimization. However, we propose leveraging language to describe visual differences more precisely, providing specific referential information at the textual semantic level. Inspired by DPO in LMs [61], we introduce the semantic-level consistency alignment by further modeling the semantic information of videos at the textual level. Specifically, for each preference video pair $y_w$ and $y_l$, we first consider the prompt $x$ as the preferred caption $c_w$. A VLM (*i.e.*, Qwen2.5-VL [4]) is then tasked to modify $c_w$ based on $y_l$, adjusting the inconsistent parts to generate $c_l$. This process ensures that $c_w$ and $c_l$ only differ in textual descriptions where the videos are inconsistent, thereby facilitating more targeted optimization. The $u_{\text{Semantic}}(c_w, c_l, y_w, t)$ within $\mathcal{L}_{\text{Semantic}}$ is then formulated as:

$$u_{\text{Semantic}} = \log \frac{p_\theta(y_w|c_w, t)}{p_{\text{ref}}(y_w|c_w, t)} - \log \frac{p_\theta(y_w|c_l, t)}{p_{\text{ref}}(y_w|c_l, t)}. \tag{8}$$

By integrating instance, state, motion, and semantic-level preference optimization, the overall loss function for PhysHPO is defined as follows:

$$\mathcal{L}_{\mathcal{P}hys\mathcal{HPO}} = \mathcal{L}_{\text{Instance}} + \lambda\mathcal{L}_{\text{State}} + \rho\mathcal{L}_{\text{Motion}} + \mu\mathcal{L}_{\text{Semantic}}, \tag{9}$$

where $\lambda$, $\rho$, and $\mu$ denote the loss weights.

## 6 Experiments

In this section, we conduct extensive experiments to answer the following research questions: (**RQ1**) Does PhysHPO enhance the physical plausibility of generated videos? (**RQ2**) Does PhysHPO compromise other performance aspects? (**RQ3**) How sensitive is PhysHPO to its key components? (**RQ4**) Is PhysHPO more advantageous than SFT for efficiency, effectiveness, and generalizability?

### 6.1 Experimental Settings

**Baselines** We apply PhysHPO to the advanced models, CogVideoX-2B and 5B [90]. Due to the unavailability of open-sourced code and model weights for SFT-based methods, *i.e.*, WISA [75] and SynVideo [98], we focus our comparisons on the following baselines: PhyT2V [CVPR'25] [85], vanilla DPO [CVPR'24] [72], SFT [90], and the respective base models. Implementation and results on HunyuanVideo [39] are shown in **Appendix §A**.

**Evaluations** To evaluate the effectiveness of PhysHPO, we adopt benchmarks focusing on two key aspects: ❶ *Physics-focused*: (*i*) VideoPhy [6] for interactions of solid-solid, solid-fluid, and fluid-fluid. (*ii*) PhyGenBench [54] for mechanics, optics, thermal, and materials. ❷ *General Capability*: VBench [34] for both overall quality and semantic.

**Implementation Details** We train base models on our selected dataset with a global batch size of $8$, using the AdamW optimizer and a learning rate of $2e-5$. Instance-level non-preferred weights are set to $\beta_{err} = 0.7$ and $\beta_{gap} = 0.3$, with $N = 2$ for state-level samples. The loss weights $\lambda$, $\rho$, and $\mu$ are set to $0.4$, $0.3$, and $0.2$, respectively. All experiments are conducted on 8 NVIDIA H100 GPUs.

### 6.2 Physical & General Performance of PhysHPO

To answer **RQ1** and **RQ2**, we comprehensively compare PhysHPO with three baseline methods on physics-focused and general benchmarks (Table 2), along with the qualitative results and user study shown in Figure 1, 4 and Figure 5 (*Left*), respectively. Our observations are summarized as follows:

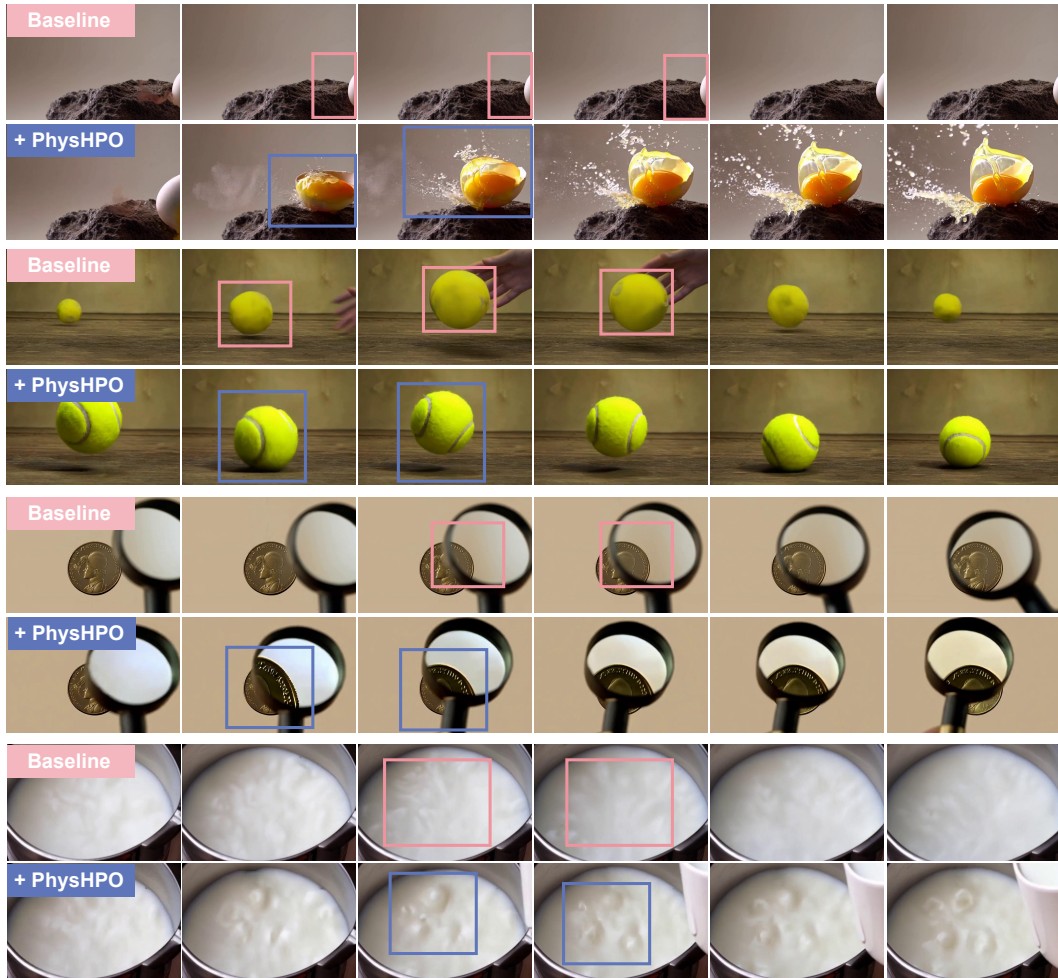

Figure 4: Qualitative results of PhysHPO. Due to space limitations, results of other baselines are provided in **Appendix** §D. Text prompts are sourced from PhyGenBench [54]. (*1st*) *Materials: A delicate, fragile egg is hurled with significant force towards a rugged, solid rock surface, where it collides upon impact.* (*2nd*) *Mechanics: A vibrant, elastic tennis ball is thrown forcefully towards the ground, capturing its dynamic interaction with the surface upon impact.* (*3rd*) *Optics: A magnifying glass is gradually moving closer to a coin, revealing the intricate details and textures of the embossed design as it approaches.* (*4th*) *Thermal: A timelapse captures the transformation of milk in a kettle as the temperature rapidly rises above $100°C$.*

**Obs.❹** *PhysHPO demonstrates superior performance in enhancing both physical fidelity and general quality.* As illustrated in Table 2, our PhysHPO consistently outperforms baseline methods (*i.e.*, PhyT2V [85] and vanilla DPO [72]), which exhibit minimal or even negative performance gains, across three physics-focus benchmarks, each targeting distinct dimensions of physical plausibility. Table 2 further highlights its robustness on VBench [34] for general capabilities. Qualitative comparisons in Figure 1 and Figure 4 provide visual confirmation of PhysHPO's ability, showcasing clear advantages over the base model. **Obs.❺** *PhysHPO effectively align video generation with human preference.* Figure 5 (*Left*) shows the user study conducted on both physical and general dimensions, where PhysHPO consistently outperforms baselines in aligning video generation with human preferences, demonstrating its superiority in preference alignment.

## 6.3 Ablation Analysis

To answer **RQ3**, we conduct evaluations on VideoPhy [6] for the contributions of PhysHPO's each level and their combinations, as shown in Table 3. We give the following observations: **Obs.❻** *Effectiveness of Hierarchical Preference Optimization.* Table 3 demonstrates a steady performance decline as $\mathcal{L}_{\text{Semantic}}$, $\mathcal{L}_{\text{Motion}}$, and $\mathcal{L}_{\text{State}}$ are progressively removed. This underscores the effectiveness

Table 3: Ablation study on level losses.

| Method | SS↑ | SF↑ | FF↑ | Over.↑ |
|---|---|---|---|---|
| **PhysHPO** | **32.4** | **54.1** | **58.9** | **45.9** |
| w/o $\mathcal{L}_{\text{Semantic}}$ | 31.7 | 53.4 | 57.1 | 45.1 |
| w/o $\mathcal{L}_{\text{Semantic}}$, $\mathcal{L}_{\text{Motion}}$ | 30.3 | 52.1 | 55.4 | 43.6 |
| Only w/ $\mathcal{L}_{\text{Instance}}$ | 28.9 | 50.7 | 51.8 | 41.9 |
| Vanilla DPO w/ Our Data | 28.2 | 50.0 | 51.8 | 41.3 |

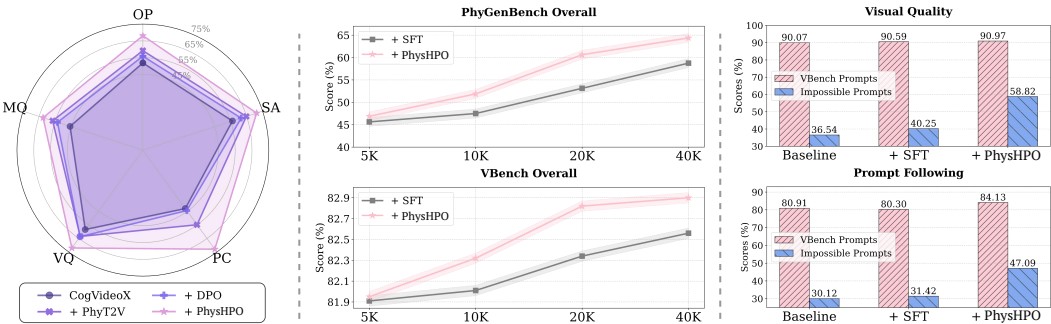

Figure 5: (*Left*) User study across five dimensions: overall preference, semantic adherence, physical commonsense, visual quality, and motion quality. (*Middle*) Performance comparison between PhysHPO and SFT under varying data volumes. (*Right*) Robustness testing with IPV-TXT [5].

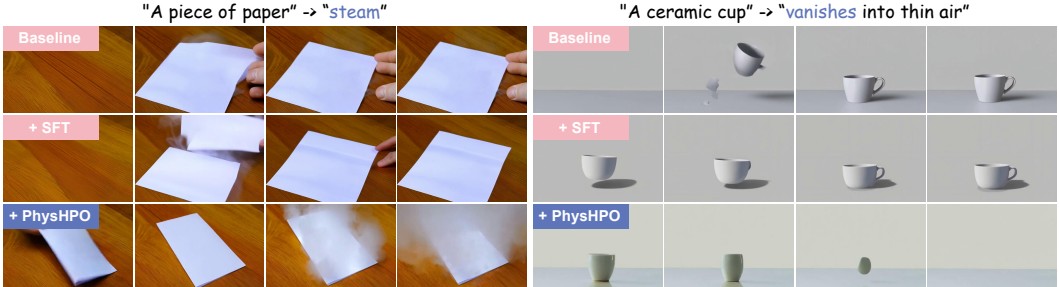

Figure 6: Robustness testing demonstration. Detailed prompts can be found in **Appendix** §C.

of the hierarchical preference optimization framework, where each level contributes uniquely to improving physical fidelity. **Obs.❼ *Importance of Fine-grained Alignment.*** Fine-grained alignment, both explicit and implicit, is critical for optimization: *i*) The removal of instance-level gap video alignment (`"Only w/ $\mathcal{L}_{\text{Instance}}$"` *vs.* `"Vanilla DPO w/ Our Data"`) highlights the importance of visually explicit modeling in bridging gaps between generated and real-world semantics and dynamics. *ii*) The exclusion of $\mathcal{L}_{\text{Semantic}}$ reveals the necessity of implicit text-video alignment for capturing nuanced and coherent relationships between textual prompts and generated videos.

## 6.4 Analysis of PhysHPO *vs.* SFT

To answer **RQ4**, we compare PhysHPO and SFT using the same data on PhyGenBench for physical fidelity and VBench for general quality, as shown in Figure 5 (*Middle*). Inspired by [5], we further assess both methods with impossible prompts (*e.g.*, "A car drives through the ocean as if it were flying") to evaluate whether improvements in physical fidelity also enable better handling of imaginative or physically impossible scenarios, with results presented in Figure 5 (*Right*) and Figure 6. We give the following observations: **Obs.❽ *PhysHPO is a data-efficient video generation enhancer.*** Figure 5 (*Middle*) demonstrates that PhysHPO consistently outperforms SFT across varying data volumes in both physics-focused and general quality metrics. This reflects PhysHPO's ability to achieve superior performance by leveraging training data more efficiently through fine-grained information utilization. **Obs.❾ *PhysHPO enables creative generalization beyond physical laws.*** As shown in Figure 5 (*Right*) and Figure 6, PhysHPO performs better on impossible prompts, generating videos that align more closely with semantic intent while maintaining internal consistency. This suggests that enhancing physical fidelity not only improves adherence to real-world physics but also equips the model with a stronger ability to generalize and adapt to imaginative scenarios, demonstrating learning beyond simple physical rules.

## 7 Conclusion

In this paper, we propose **PhysHPO**, a novel framework for Hierarchical Cross-Modal Direct Preference Optimization, enhancing video fine-grained alignment across four levels: instance, state, motion, and semantic. Recognizing real-world videos as the best reflections of physical phenomena, we introduce an automated data selection pipeline to efficiently utilize large-scale text-video datasets, removing the need for exhaustive dataset construction. Extensive evaluations on both physics-focused and general benchmarks demonstrate that PhysHPO significantly improves the physical plausibility and overall quality of existing video generation models, addressing key challenges in physically plausible video generation.

## Acknowledgement

This research was supported by the Innovation and Technology Fund of HKSAR under grant number GHX/054/21GD.

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

# A  More Results on HunyuanVideo

To further validate the superiority of our proposed PhysHPO, in this section, we further apply PhysHPO on HunyuanVideo-540p [39] implemented by FastVideo [97].

| Method | VideoPhy [6] | | | | PhyGenBench [54] | | | | | VBench [34] | | |
|---|---|---|---|---|---|---|---|---|---|---|---|---|
| | SS↑ | SF↑ | FF↑ | Over.↑ | Mechanics↑ | Optics↑ | Thermal↑ | Materials↑ | Overall↑ | Total↑ | Quality↑ | Semantic↑ |
| HunyuanVideo [39] | 19.7 | 43.2 | 42.9 | 33.4 | 37.5 | 58.0 | 36.7 | 45.0 | 45.6 | 81.4 | 83.1 | 74.9 |
| + PhyT2V [85] | 21.1 | 45.9 | 50.0 | 36.3 | 40.0 | 58.0 | 33.3 | 47.5 | 46.3 | 80.5 | 82.0 | 74.5 |
| **+ PhysHPO (Ours)** | **26.8** | **50.7** | **55.4** | **41.6** | **47.5** | **62.0** | **43.3** | **60.0** | **54.4** | **81.9** | **83.5** | **75.7** |

Table 4: Evaluation of PhysHPO on HunyuanVideo [39].

**Quantitative Results**  Table 4 presents the quantitative results of PhysHPO on HunyuanVideo. Consistent with our **Obs.❹** in the main content, PhysHPO achieves superior performance in enhancing both physical fidelity and overall video quality. These results further validate its effectiveness.

**Qualitative Results**  We demonstrate results in Figure 7. More results on *human action* are shown in Section §D.

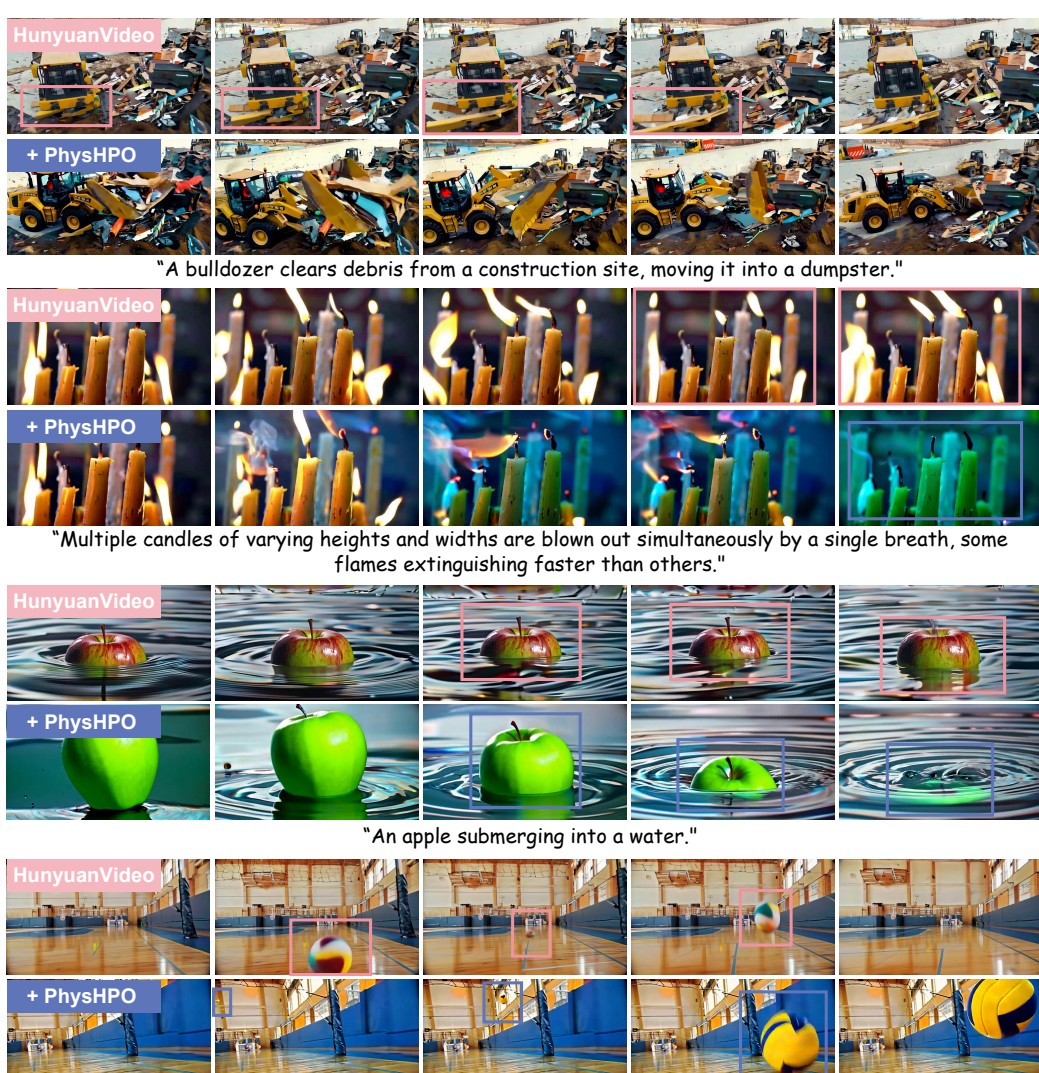

"A bulldozer clears debris from a construction site, moving it into a dumpster."

"Multiple candles of varying heights and widths are blown out simultaneously by a single breath, some flames extinguishing faster than others."

"An apple submerging into a water."

"A volleyball is spiked, hitting the wooden floor of the court and producing a soundless bounce visible by its slight dip and subsequent rebound."

Figure 7: Qualitative demonstration of PhysHPO on HunyuanVideo [39].

# B   More Details of Data Selection

**In-depth Evolving Prompting**   To achieve more fine-grained scoring for the physical fidelity of real-world videos, we employ the in-depth evolving prompting strategy [84]. This approach augments sample captions to emphasize specific physical phenomena. An example of our crafted *evolving-dynamic* prompt is shown in Figure 8.

```
                              Evolving Dynamic Prompt

prompt: '''You are an expert in physics and natural language processing, specializing in analyzing and enhancing textual
descriptions based on physical phenomena. Your task is to process a given {caption} and perform the following steps.

### Step 1: Enhance the caption based on six physical phenomena
For the given {caption}, enhance it to emphasize each of the following six dynamic phenomena. The enhancement must strictly be
based on the original caption's content. Do not invent new events, objects, or scenarios. Instead, focus on rephrasing and
elaborating on the existing information in the caption to highlight the specific characteristics of each phenomenon. Ensure
that the enhanced captions are vivid, precise, and aligned with the corresponding physical behavior:

1. Rigid body motion: Focus on the movement of solid objects that maintain their shape and structure, such as rotation or
translation.
2. Collision: Emphasize interactions where two or more objects come into contact and exchange momentum or energy.
3. Liquid motion: Highlight any aspects of the caption that involve fluid-like behaviors, such as flow, splashing, or other
liquid characteristics.
4. Gas motion: If the caption involves air or gas, describe phenomena such as diffusion, expansion, or turbulence. If no gas
motion is mentioned, rephrase the caption to make any relevant gaseous interactions clearer without adding new elements.
5. Elastic motion: Focus on stretching, compressing, or oscillatory motion of elastic materials if present in the caption.
Elaborate on any such motions to make them more vivid.
6. Deformation: Highlight changes in shape or structure of materials under stress, such as bending, twisting, or breaking.

Output six enhanced captions, each tailored to one of the above phenomena. Ensure that all enhancements remain faithful to the
original caption's content.

### Step 2: Score the enhanced captions
After generating the six enhanced captions, evaluate each caption based on how well it represents the corresponding dynamic
phenomenon. Provide a score from 1 to 5 (1 = poor, 5 = excellent) for each caption. Additionally, provide a brief reason for
the score, explaining why the caption is effective or where it could be improved.

### Final Output Format:
Your output should follow this exact format:
Caption of rigid body motion: [Enhanced caption]; Score: [1-5]; Reason: [Explanation].
Caption of collision: [Enhanced caption]; Score: [1-5]; Reason: [Explanation].
Caption of liquid motion: [Enhanced caption]; Score: [1-5]; Reason: [Explanation].
Caption of gas motion: [Enhanced caption]; Score: [1-5]; Reason: [Explanation].
Caption of elastic motion: [Enhanced caption]; Score: [1-5]; Reason: [Explanation].
Caption of deformation: [Enhanced caption]; Score: [1-5]; Reason: [Explanation].

### Example:
...

'''
```

Figure 8: Demonstration of crafted *evolving-dynamic* prompt.

Similarly, the *evolving-thermodynamic* and *evolving-optic* prompts follow the same structure.

In addition to the performance comparison between our evolving-based scoring and direct scoring shown in Figure 3, we also provide examples with detailed scores under different strategies in Figure 9.

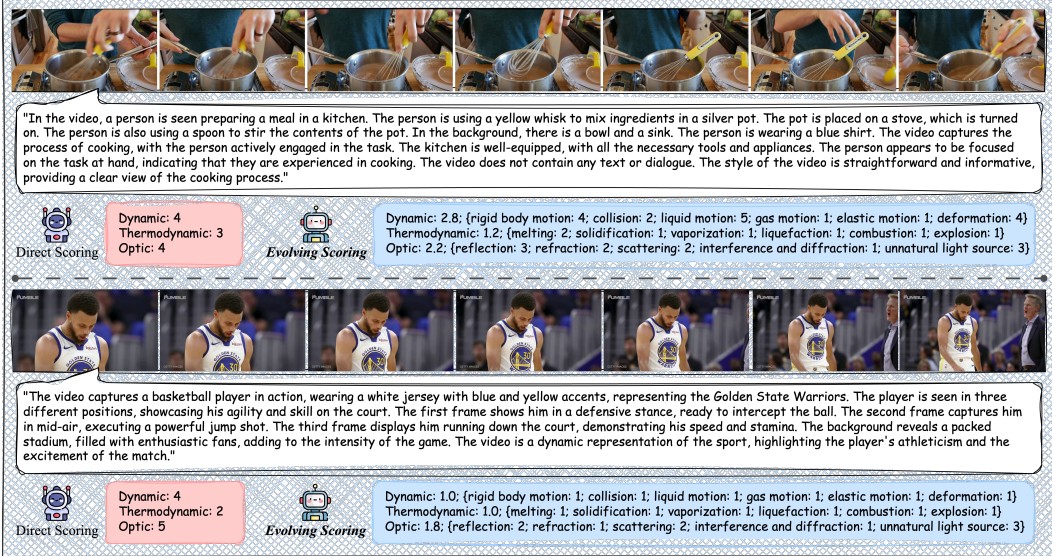

Figure 9: Comparison of direct scoring *vs.* our evolving scoring.

Specifically, we compare a sample of *dynamic* stirring (*Top*) with another sample where the camera zooms out but the scene remains *static* (*Bottom*). As shown, the direct scoring strategy assigns similar scores to these two distinct videos, while our evolving-based scoring provides more fine-grained differentiation and accurate assessments.

**Analysis of Caption-based Selection**    In our data selection process, we first performed *video-based* filtering to ensure reality, followed by *caption-based* filtering to enhance physical fidelity and diversity. Here, we further validate the efficiency and effectiveness of our caption-based selection strategy.

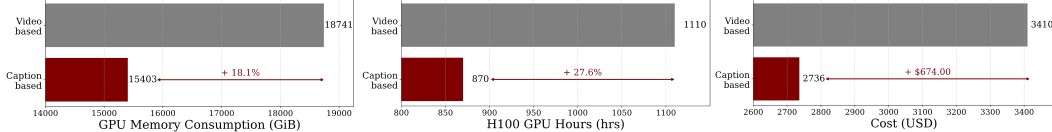

Figure 10: Efficiency comparison of video-based selection *vs.* our caption-based selection. (***Left***) GPU Memory Consumption (GiB). (***Middle***) H100 GPU Hours (hrs). (***Right***) Cost (USD).

Figure 10 first demonstrates an efficiency comparison of our caption-based selection and video-based selection. Specifically, ❶ GPU Memory Consumption (*Left*): Caption-based selection requires significantly less GPU memory, reducing usage by $18.1\%$ compared to video-based selection ($15,403$ GiB *vs.* $18,741$ GiB). This reduction highlights the computational efficiency of our method in terms of memory footprint. ❷ H100 GPU Hours (*Middle*): Video-based selection increases compute time by $27.6\%$ ($3,410$ hrs *vs.* $2,736$ hrs), indicating that caption-based selection is more time-efficient and better suited for large-scale processing. ❸ Cost (*Right*): At scale, video-based workflows incur an additional $674.00 in cost compared to caption-based workflows, further emphasizing the cost-efficiency of our approach.

To further validate the effectiveness of our caption-based selection, we randomly sampled 100 data points from the reality-selected data pool for human scoring. We then analyze the scoring accuracy by comparing human scoring results with three scoring methods: video-based scoring, caption-based direct scoring, and our caption-based evolving scoring. As demonstrated in Figure 11, our caption-based evolving scoring achieves the highest alignment with human scoring across all three evaluated dimensions. These results further confirm that our method not only improves computational efficiency but also ensures superior scoring reliability.

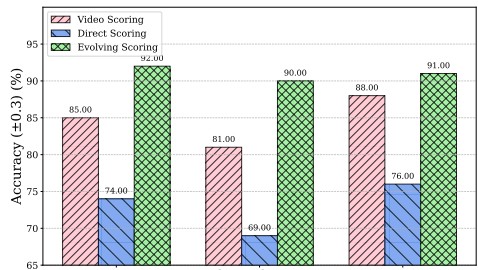

Figure 11: Adjacency accuracy ($\pm 0.3$) of different scoring strategies with human scoring.

**Statistical Overview**    To provide a clearer understanding of each stage in our data selection process, we present the statistical breakdown in Table 5. The raw data is sourced directly from OpenVidHD-0.4M [57]. Moving forward, we will process more existing high-quality text-video datasets (*e.g.*, MiraData [37], InternVid [76]) using our data selection pipeline to facilitate future research.

Table 5: Statistics of data selection.

| Data Selection | Total Count |
|---|---|
| Raw Data | 433,523 |
| + Reality | 266,931 |
| + Physics | 59,076 |
| + Diversity | 21,085 |

**Data Sample Demonstration**    Real-world videos serve as direct reflections of physical phenomena. Here, we present examples of our selected videos as illustrative examples, as shown in Figure 12. These examples highlight diverse physical scenarios, providing a reference for the types of phenomena captured in our dataset.

## C    More Experimental Settings and Analysis

### C.1    More Details of Experimental Settings

**Physics-focused Metric**    We evaluate the physical fidelity of generated videos using two widely adopted physics-focused benchmarks: VideoPhy [ICLR'25] [6] and PhyGenBench [ICML'25] [54].

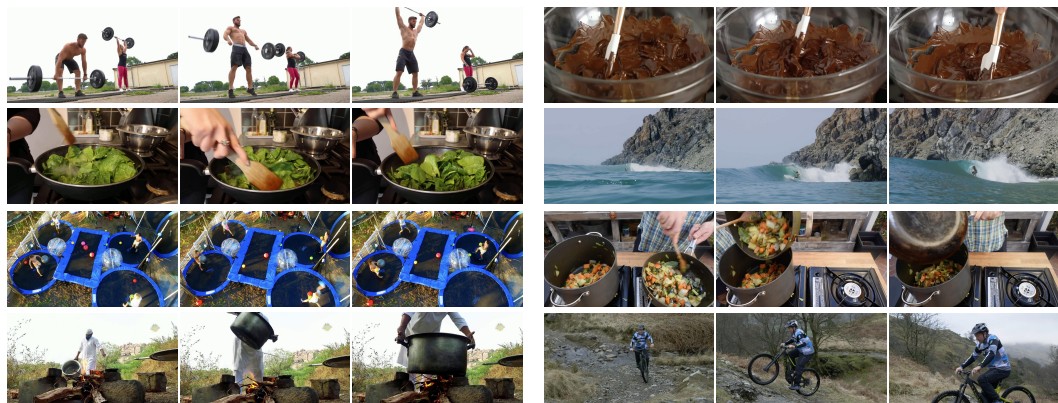

Figure 12: Randomly sampled data demonstration from our selected dataset.

Following the evaluation protocol introduced in WISA [75], we leverage VideoCon-Physics from VideoPhy [6] to assess two key metrics: **Physical Law Consistency (PC)** and **Semantic Coherence (SA)** of the generated videos. Specifically, we use 344 carefully curated prompts from VideoPhy and 160 prompts from PhyGenBench, both designed to reflect a diverse range of physical principles and scenarios.

To quantify performance, we follow the scoring criteria outlined in the respective benchmark papers. For both VideoPhy and PhyGenBench, we consider PC and SA values greater than or equal to 0.5 as PC = 1 and SA = 1, while values less than 0.5 are treated as PC = 0 and SA = 0. Specifically:

- For **VideoPhy**, we report the proportion of generated videos where both PC = 1 and SA = 1, reflecting the alignment with physical laws and semantic intent simultaneously.

- For **PhyGenBench**, we focus exclusively on the Physical Commonsense Alignment (*PCA*) score, which evaluates the consistency of physical reasoning under the condition that PC = 1.

This evaluation framework ensures consistency with prior work and provides a robust assessment of the physical fidelity and semantic coherence of the generated videos.

**Impossible Robustness Testing**   Inspired by Impossible Videos [ICML'25] [5], we further employ prompts from IPV-TXT [5] to evaluate whether PhysHPO goes beyond merely fitting fixed physical phenomena and demonstrates stronger generalization and robustness. The results are illustrated in Figure 5 (*Right*) and Figure 6.

For the quantitative comparison shown in Figure 5 (*Right*), we follow the evaluation protocol in [5] to assess the generated videos on two key metrics:

- **Visual Quality (VQ):** This metric is derived by combining six factors from VBench [34], including Subject Consistency, Background Consistency, Motion Smoothness, Aesthetic Quality, Imaging Quality, and Dynamic Degree. These factors are aggregated into a single score to reflect the overall visual quality of the generated videos.

- **Impossible Prompt Following (IPF):** This metric evaluates how well the generated videos align with the semantic intent of the impossible prompts. Following [5], we utilize GPT-4o to provide a binary judgment for each video based on prompt adherence and calculate the proportion of positive judgments as the final IPF score.

For the qualitative comparison presented in Figure 6, we further detail the prompts here:

- *Left*: *"A piece of paper mysteriously transforms into steam on a wooden table. The surreal sequence begins with someone placing the paper down, followed by a gentle touch from a human hand that triggers an unexpected reaction - the solid paper instantly vaporizes into wisps of white steam that dissipate into the air."*

- *Right*: *"A ceramic cup sitting on a plain table surface mysteriously vanishes into thin air in this photo-realistic footage. The abrupt disappearance happens without any visible cause, defying physics in an uncanny way. The scene is captured in crisp, lifelike detail with natural lighting."*

**User Study**   We conduct a user study to evaluate human preferences using both the mean opinion score (MOS) and direct pairwise comparisons. Specifically, we design a user-friendly interface to

facilitate the evaluation process and collect feedback from a total of 15 volunteers. The detailed instructions provided to participants are shown below.

---

**User Study: Physically Plausible Video Generation**

Thank you for participating in our user study! Please follow these steps to complete your evaluation:

1. **Video Generation:** Carefully read the target prompt provided, and then view the provided videos.

2. **Scoring Criteria:** Assign a score to each generated video based on the following aspects (1 being the lowest, 5 being the highest):

- *Overall Preference*: A holistic evaluation of your overall satisfaction with the generated video, including aspects such as semantics, physical reasoning, visual quality, and motion quality.
- *Semantic Adherence*: How accurately the video reflects the input semantic description or textual instructions.
- *Physical Commonsense*: Whether the video content aligns with basic physical commonsense, such as the laws of motion, object interactions, and logical behavior.
- *Visual Quality*: The visual appearance of the video, including resolution, clarity, color representation, and texture details.
- *Motion Quality*: The smoothness and naturalness of motion in the video, including object or character trajectories and speed variations.

3. **Submission:** Click the "Submit Scores" button to submit your scores.

**Notations:**
1. We observe that the edge browser is not fully compatible with our interface. Chrome is recommended.
2. Remember to click the "Submit Scores" button after your evaluation.
3. If you see that videos and the score sliders are not aligned, shrinking your page usually works.
4. If the video seems to be stuck, usually waiting for a few seconds will sovle this.
5. If the page is not responsive for a long time, please try to refresh it.
6. If you have any questions, please directly contact us. Thank you for your time and effort!

---

## C.2 More Analysis

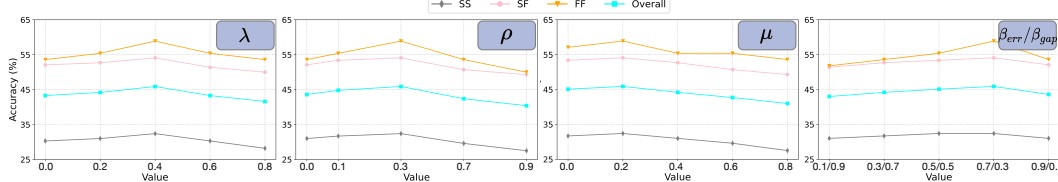

Figure 13: Hyperparameter analysis of PhysHPO on VideoPhy [6].

**Analysis of Hyperparameters** Figure 13 presents a detailed analysis of the impact of various hyperparameters on the performance of PhysHPO on the VideoPhy [6] benchmark. *a*) The first plot examines the state-level loss weight ($\lambda$), showing that accuracy improves as $\lambda$ increases, peaking at $\lambda = 0.4$, before declining. This suggests that moderate emphasis on state-level consistency enhances performance, while excessive weighting may lead to overfitting to low-level details. *b*) Similarly, the motion-level loss weight ($\rho$) in the second plot demonstrates a comparable trend, with performance reaching its best at $\rho = 0.3$. This highlights the importance of capturing motion dynamics, but also the potential detriment of overemphasizing this aspect. *c*) The third plot explores the semantic-level loss weight ($\mu$), where accuracy improves with increasing $\mu$ up to $\mu = 0.2$, after which the performance plateaus or slightly declines. This underscores the necessity of aligning semantic-level information to ensure coherent and contextually accurate video generation. *d*) Finally, the fourth plot analyzes the balance between error ($\beta_{\text{err}}$) and gap ($\beta_{\text{gap}}$) samples in instance-level negative sampling. Results

indicate that the weighting ($\beta_{\text{err}}/\beta_{\text{gap}} = 0.7/0.3$) achieves the highest accuracy, as both sample types contribute to model robustness. Overemphasizing either error ($\beta_{\text{err}}/\beta_{\text{gap}} = 0.9/0.1$) or gap samples (0.1/0.9) diminishes performance, likely due to an imbalanced training signal.

**Analysis of Non-preferred Samples** The choice of non-preferred samples determines their contrast with preferred samples and significantly impacts the model's focus on preference learning. Here, we further explore the selection strategies for non-preferred samples at the instance, state, and motion levels, while ensuring consistency with the corresponding preferred sample configurations in the main text. Specifically, *i*) **Instance-level:** The number of generated error videos corresponding to a preferred video, *i.e.*, how many videos are generated to select the error video; *ii*) **State-level:** The number of boundary frames swapped to construct non-preferred samples; *iii*) **Motion-level:** The choice of structural representations, *e.g.*, depth map, optical flow, and Canny edge.

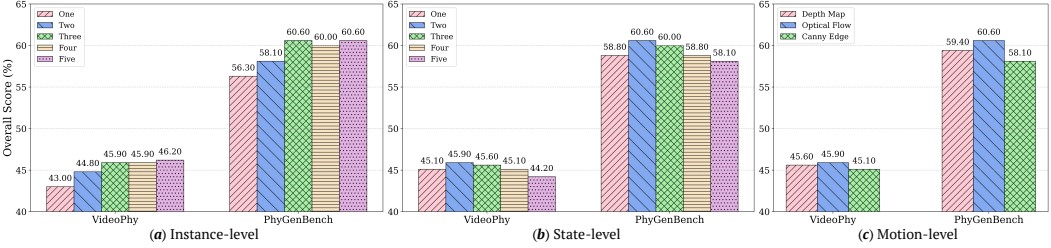

Figure 14: Non-preferred samples analysis of (*a*) Instance-level, (*b*) State-level, and (*c*) Motion-level.

Figure 14 illustrates the impact of different non-preferred sample selection strategies on model performance: (*a*) Instance-level: We observe that selecting three generated samples achieves the best performance. While increasing the number of samples can lead to slight performance gains, it also incurs higher computational costs, making three a practical balance. (*b*) State-level: Swapping a moderate number of boundary frames yields optimal results, as excessive or insufficient frame swapping diminishes the contrast between preferred and non-preferred samples. (*c*) Motion-level: Optical flow achieves the highest overall score, demonstrating its effectiveness in capturing motion dynamics. In contrast, depth maps and Canny edge representations show relatively lower performance, likely due to their less detailed motion information. These results emphasize the importance of carefully balancing computational efficiency and performance when selecting non-preferred samples. Proper configurations at each level ensure effective and robust model performance.

## D  Exhibition Board

We provide more comparison results here in Figure 15 (with baselines) and Figure 16 (with physically focused prompts), along with results on ***human action/motion*** in Figure 17 (HunyuanVideo [39]) and Figure 18 (CogVideoX [90]). *We would highly recommend watching the **Webpage**.*

## E  Limitation and Future Works

While PhysHPO excels at aligning videos with fine-grained precision, the substantial computational cost required for training large-scale video generation models remains a potential limitation, particularly for individuals and organizations with limited resources. Future research should explore more lightweight architectures (*e.g.*, FramePack [96]) or further explore test-time preference alignment to achieve greater performance improvements with reduced computational overhead. Additionally, the data selection strategy introduced in this paper is designed to be relatively simple for practicality. Future work should further optimize it based on specific task requirements.

## F  Ethical Implications

PhysHPO is developed as a hierarchical preference optimization strategy for RESEARCH ONLY. It may still raise important ethical considerations, particularly around content generation. The ability to generate high-quality videos can potentially be misused for creating misleading or harmful content. To mitigate this risk, we recmmend incorporating safeguards such as adding watermarks to generated videos to ensure transparency and authenticity. Additionally, guidelines on responsible use should be established, emphasizing its application in ethical and creative contexts, such as educational, artistic, or research-based scenarios, while discouraging its use for deceptive or harmful purposes.

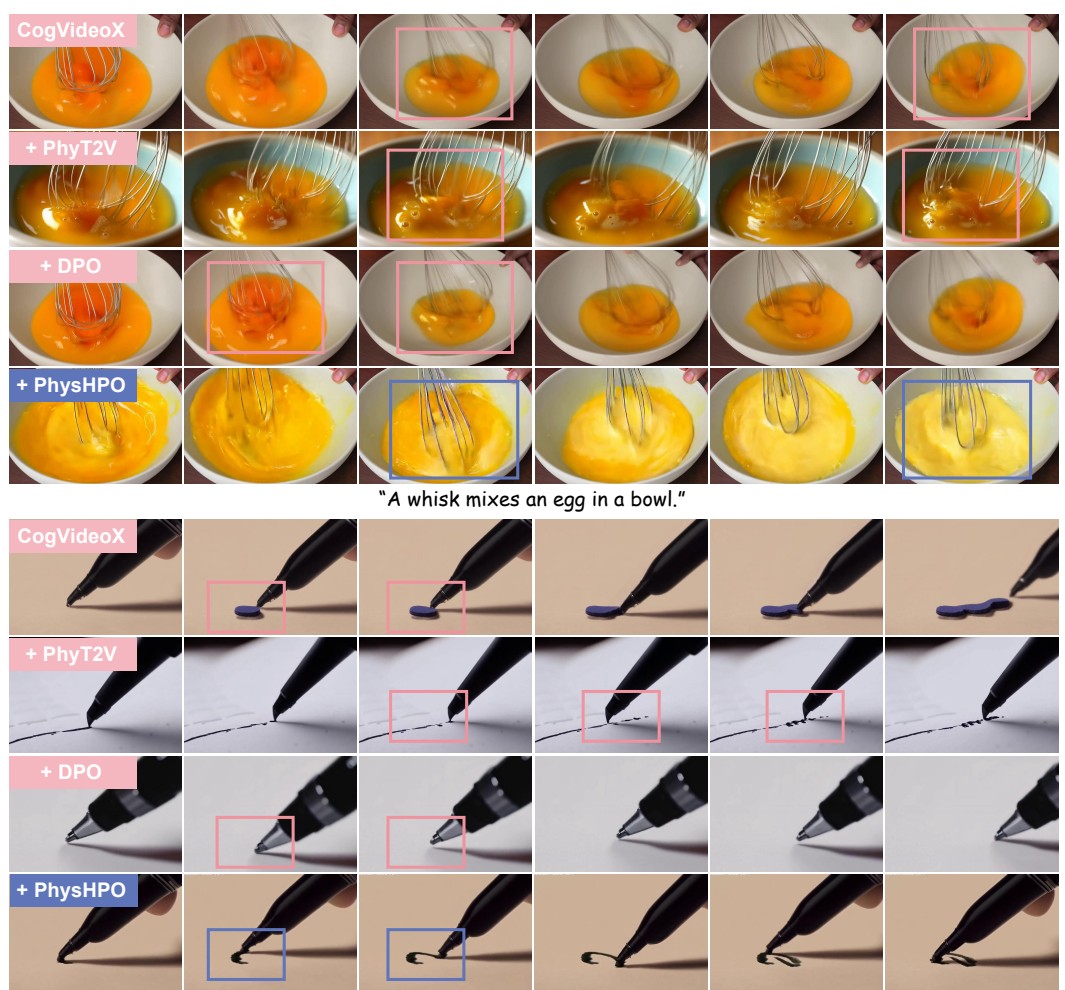

"A whisk mixes an egg in a bowl."

"A black pen is used to write on the smooth, white surface of a notebook, showcasing the interaction between the pen and the notebook surface."

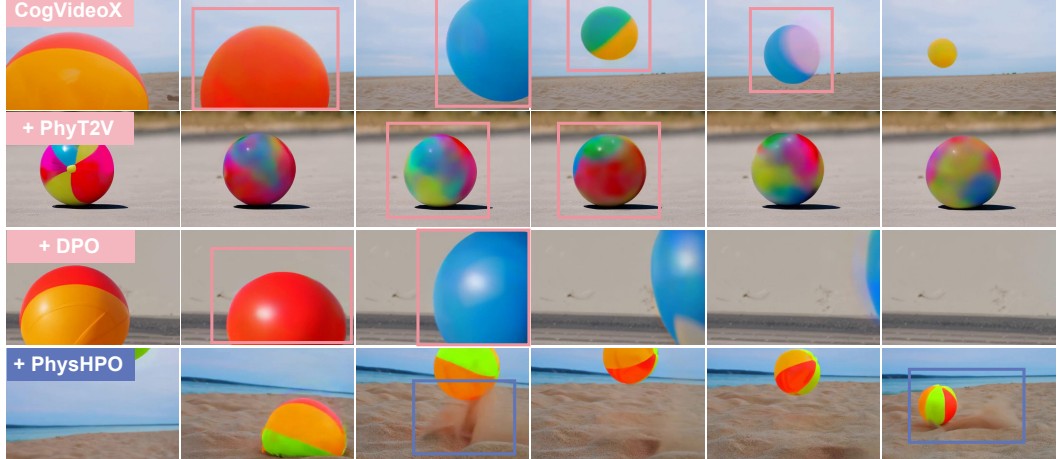

"A vibrant, elastic beach ball is thrown forcefully towards the ground, capturing its dynamic interaction with the surface upon impact."

Figure 15: More comparison demonstrations with baselines.

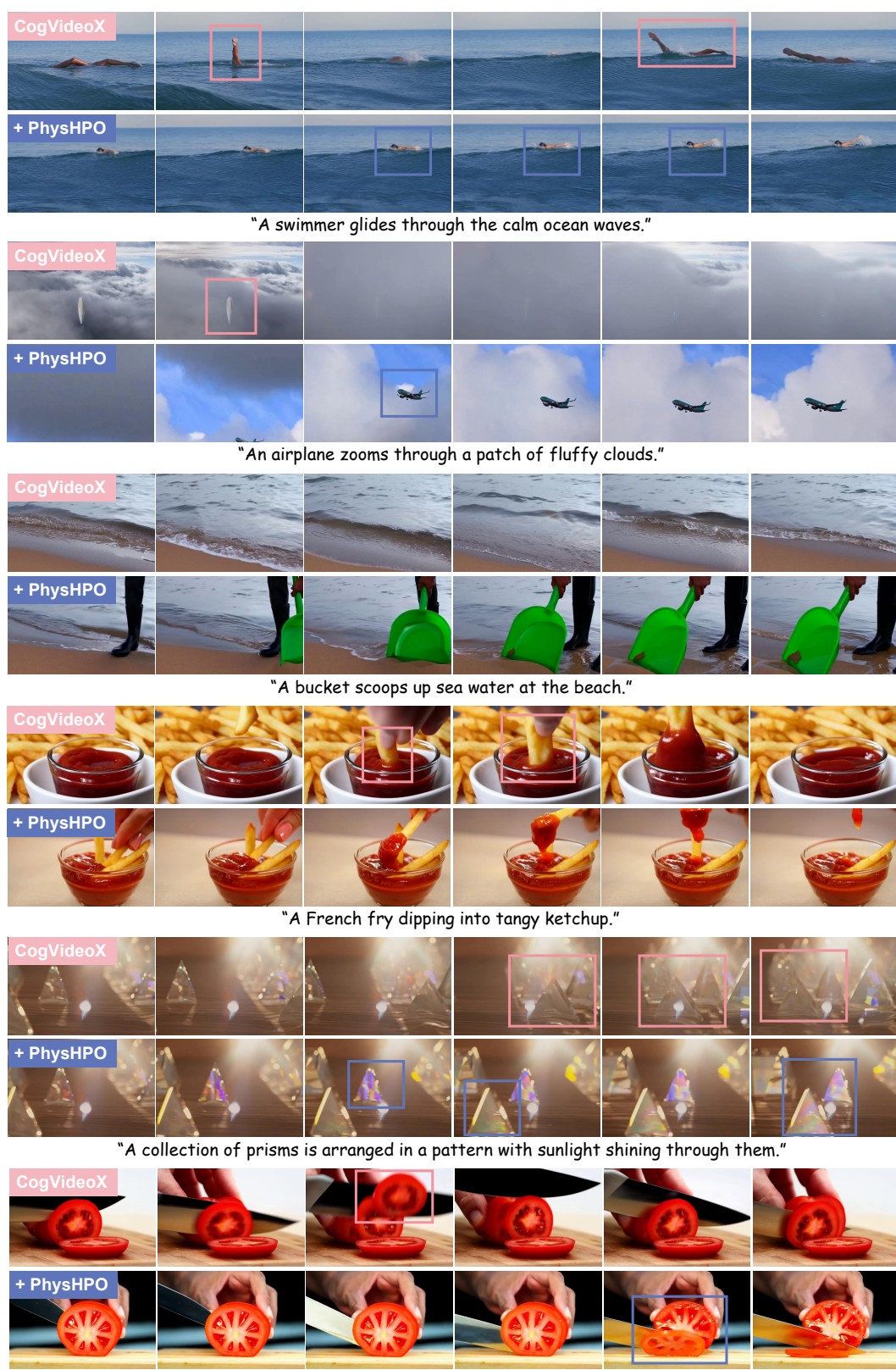

Figure 16: More results demonstrations with prompts sourced from VideoPhy [6].

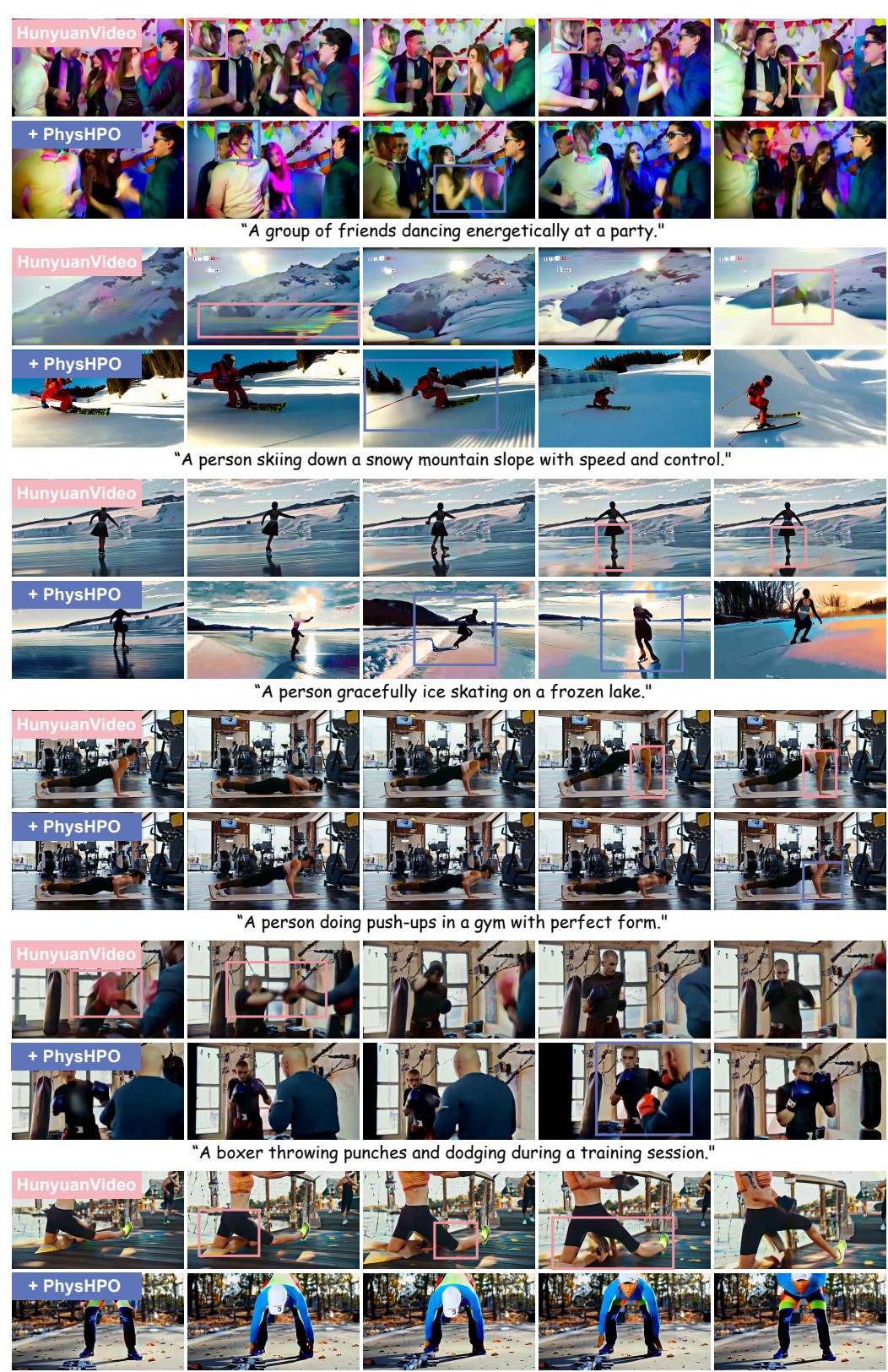

Figure 17: More results demonstrations with human action/motion-focused prompts.

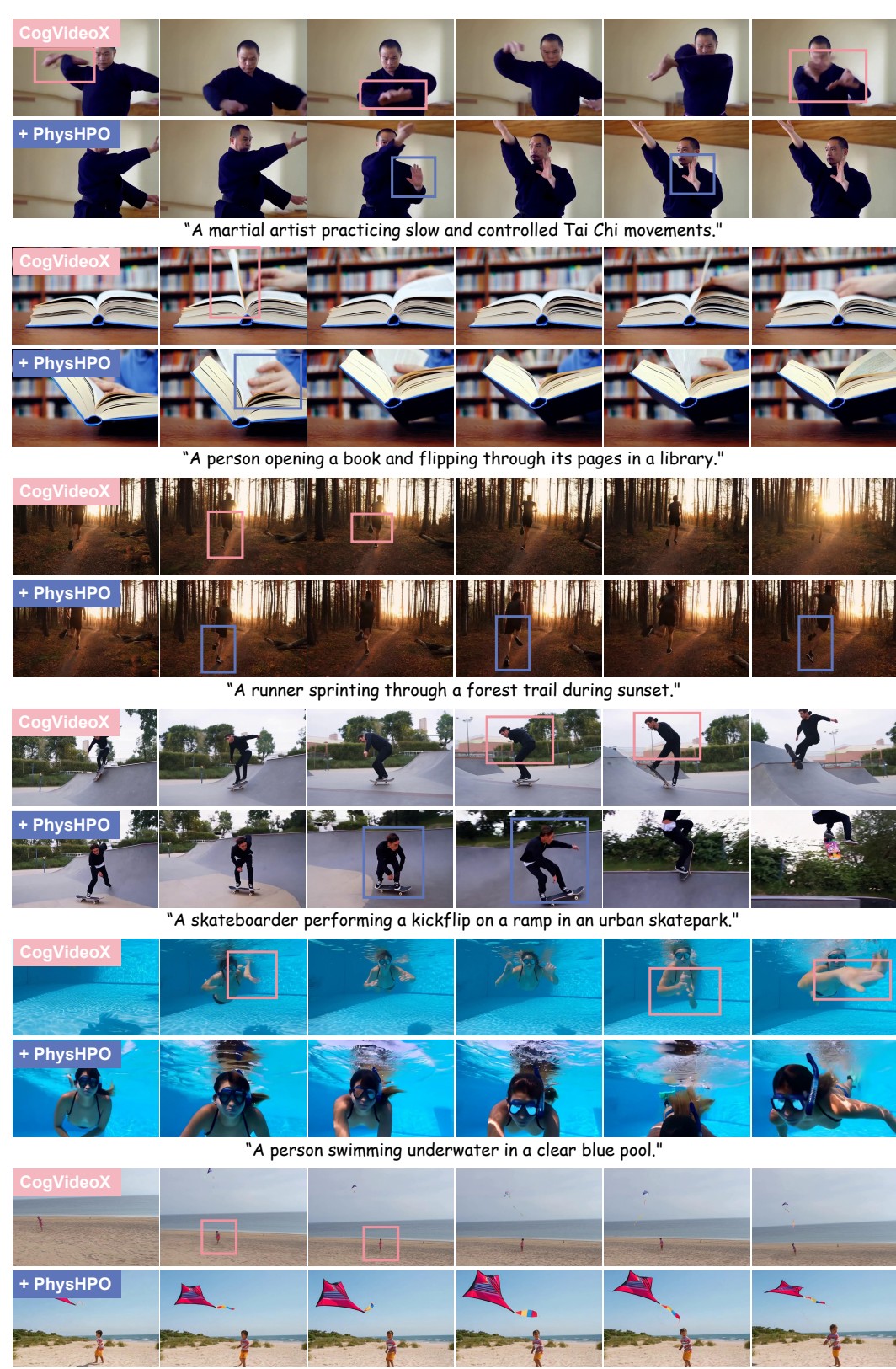

Figure 18: More results demonstrations with human action/motion-focused prompts.

