# OpenReview forum: "Hierarchical Fine-grained Preference Optimization for Physically Plausible Video Generation"
_NeurIPS.cc/2025/Conference — NeurIPS 2025 poster_

### Official Review · Reviewer_JHEX · 2025-07-03

**Clarity:** 2
**Significance:** 3
**Originality:** 3
**Rating:** 4
**Confidence:** 3

**Summary:**

The paper introduces PhysHPO, a framework for improving the physics of T2V model using a fine-grained preference optimization approach. PhysHPO includes preference pairs at different aspects, including instance-level, state-level, motion-level and semantic-level. The work also includes a data selection pipeline with the aim to select a subset of videos from a large set (~0.4M) that optimizes the fine-tuning performance. The proposed PhysHPO seems different from general DPO setting as it creates win-lose pairs by intervening the generation process of the losing video to make it worse. Experimental results show that PhysHPO outperforms a few baselines including SFT, DPO and PhyT2V.

**Questions:**

See weaknesses

**Ethical Concerns:**

["NO or VERY MINOR ethics concerns only"]

**Final Justification:**

The authors have addressed most of my concerns. Please work on the writing to resolve the confusion in notations.

**Limitations:**

See weaknesses

**Quality:**

3

**Strengths And Weaknesses:**

(+) The paper addresses fine-grained DPO for video generation, which is an important direction. The framework proposes DPO design from four different perspectives and use their ensemble to enhance physics in the generated videos.

(+) PhysHPO is essentially an annotation-free method that doesn't require human preference data. The preference label comes from either intervening the ground truth video to create a deteriorate one or by manipulating the input condition and generate a deteriorate one with the T2V model.

(+) Experiment results shows obvious improvements from PhysHPO compared to other baselines.

(-) I am not sure if this PhysHPO is actually a DPO method because it has a different setting from the original definition of DPO. There should be a paragraph that explicitly clarifies the difference between PhysHPO and DPO. I am pretty confused after reading the whole paper and if I understand correctly, PhysHPO has a different definition of p_ref() because this p_ref() is essentially the ground truth video which is assumed to always win the comparison.

(-) As suggested above, the whole writing need some improvement to make things clear. The data selection section has a very loose connection with the actual methodology section unless one could realize in advance that this is not a standard DPO method.

(-) It is not very clear to me why PhysHPO would work since it always assume the ground truth video wins the comparison and the losing video is generated by manipulating the generation process. Some of the operation seems pretty heuristic like replacing the start and ending frames with irrelevant ones (state-level), masking the prompt tokens (instance-level), rephrasing the input prompts (semantic-level).

(-) A lot the details need to be explained: what is u_motion in line 248? How is y_w actually used since it's a ground truth video instead of one being generated by the reference model? Did you add noise to it and then input the noisy ground truth video to reference model and obtain the denoising outputs as the signal for loss calculation? I feel like the equations need to be write explicitly like in the original diffusion DPO paper (e.g. eq 14). Perhaps I have misunderstood something from the paper so please clarify. 3) Since human preference annotation were not collected, how did you implemented vanilla DPO in table 2? Is the winning sample still assumed to be the ground truth video? What are the losing samples and how did you get them?

---

> ### Author Rebuttal · Authors · 2025-07-31
>
> Thank you to all the reviewers for your thoughtful and constructive comments! We are truly encouraged to see that the reviewers appreciate several positive aspects of our work, such as **strong performance** (Reviewer `epiP`, `EaEb`, `JHEX`, `R9Wj`), **comprehensive & convincing experiments** (Reviewer `epiP`, `R9Wj`), **novel & effective framework** (Reviewer `R9Wj`, `EaEb`), **valuable & practical data selection pipeline** (Reviewer `R9Wj`, `EaEb`, `JHEX`), **well-motivated methodology** (Reviewer `epiP`, `JHEX`), and **clear writing** (Reviewer `epiP`).
>
> Your thoughtful review has been invaluable in helping us enhance our manuscript. Here, we address your comments thoroughly to ensure clarity and accuracy.
>
> ---
> > `Weakness 1`: How does PhysHPO align with the original definition of DPO, given its different setting? Specifically, how is $p_{ref}()$, defined as the ground truth video that always wins comparisons, consistent with DPO?
>
> Thank you for raising this important point! We would like to respectfully clarify that PhysHPO indeed belongs to the DPO paradigm. In PhysHPO, $p_\theta$ represents the policy video diffusion model, and $p_{ref}$ denotes the reference model, as outlined in the `Preliminaries` section of the paper. The key modification in PhysHPO lies in constructing **new** positive and negative sample pairs to explore physical commonsense. This does **not** alter the standard selection process between $p_\theta$ and $p_{ref}$, ensuring PhysHPO remains consistent with the core DPO methodology.
>
> ---
> > `Weakness 2`: How can the writing be improved to better connect the data selection section with the methodology, especially given that this is not a standard DPO method?
>
> Thank you for the feedback! We will revise the writing to better clarify how the data selection ties into the methodology and explicitly highlight the connection to PhysHPO as a DPO adaptation. This will improve the flow and make the methodology clearer.
>
> ---
> > `Weakness 3`: Why does PhysHPO work when it assumes the ground truth video always wins and the losing video is generated through heuristic manipulations, such as replacing frames (state-level), masking prompt tokens (instance-level), or rephrasing prompts (semantic-level)?
>
> Thank you for your comment! The design of PhysHPO assumes that the ground truth video always wins the comparison, while losing videos are generated by introducing physical or semantic inconsistencies. Operations at instance, state, and semantic level are designed to **simulate common errors** in physical reasoning or semantic alignment. Specifically,
>
> - **State-level** (*e.g.*, replacing frames) disrupts physical continuity, mimicking unrealistic state transitions such as sudden object disappearance or appearance.
> - **Instance-level** (*e.g.*, masking prompt tokens) introduces semantic deviations, helping the model learn the relationship between the prompt and the video.
> - **Semantic-level** (*e.g.*, rephrasing prompts) creates subtle semantic inconsistencies, further challenging the model to align video generation with the intended meaning.
>
> Although these designs may appear heuristic, they are grounded in physical commonsense and semantic reasoning principles.
>
> Additionally, beyond the ablation studies in `Table 3`, which remove each level alignment progressively, we conducted further ablation experiments focusing on the contribution of each level:
>
> *Table A: Additional Ablation Study on Level Losses on VideoPhy*
> ||SS|SF|FF|Overall|
> |-|-|-|-|-|
> |Vanilla DPO w/ Our Data|28.2|50.0|51.8|41.3|
> |+ $L_\text{Instance}$|28.9|50.7|51.8|41.9|
> |+ $L_\text{State}$|29.6|50.0|53.6|42.2|
> |+ $L_\text{Motion}$|30.3|50.7|55.4|43.0|
> |+ $L_\text{Semantic}$|28.9|50.0|53.6|41.9|
>
> The results show that every level of preference alignment contributes to improving the physical plausibility of video generation. We hope this further clarifies our rationale.
>
>
> ---
> > `Weakness 4`: A lot the details need to be explained: **(1)** What is $u_{motion}$ in line 248? **(2)** How is $y_w$ (ground truth video) used, given it isn’t generated by the reference model? Did you add noise and use denoising outputs for loss calculation? **(3)** Should the equations be explicitly written, like in the original diffusion DPO paper (*e.g.*, eq. 14)? **(4)** Without human preference annotations, how was Vanilla DPO implemented in Table 2? Was the winning sample assumed to be the ground truth video, and how were the losing samples generated?
>
> Thank you for your detailed questions and suggestions, and we apologize for any potential confusion caused!
>
> 1. $u_{motion}$ refers to the motion-level loss calculation function for positive and negative video structure pairs.
>
> 2. Regarding $y_w$, it is indeed the ground truth video, but we **add noise** to it before inputting it into the reference model. This approach aligns with prior works like VideoDPO[1], where winning videos are **pre-generated** by the reference model and then used as training data, rather than being directly generated during training.
>
> 3. We appreciate your suggestion on improving the clarity of the equations and will revise the formulas in future versions to ensure they are more explicit.
>
> 4. For Vanilla DPO in `Table 2`, we implemented it **using our dataset** (*i.e.*, `"Vanilla DPO w/ Our Data"`) for fair comparison. In this setup, the winning sample corresponds to $y_w$ (ground truth video), while the losing sample corresponds to $y_l^{err}$, which is generated using the caption associated with $y_w$ as the prompt.
>
> ---
> Thank you again for your detailed feedback. We hope this rebuttal clarifies the points raised and addresses your concerns. We are happy to continue discussing ways to improve the paper.
>
> ---
> [1] VideoDPO: Omni-Preference Alignment for Video Diffusion Generation [CVPR’25]

---

### Official Review · Reviewer_EaEb · 2025-07-03

**Clarity:** 2
**Significance:** 3
**Originality:** 3
**Rating:** 5
**Confidence:** 3

**Summary:**

The paper presents an approach named PhysHPO for physically plausible video generation. It presents a cross-modal (text-video), direct preference optimization framework to enable the physical plausibility in video generation. It ensures the video alignment of the text prompt and a physically plausible video on four granular/fine-grained levels - instance, state, motion, and semantic. Additionally, it proposes an automated data selection pipeline based on reality, physical fidelity, and diversity to curate real-world video data that shows physical laws (dynamic objects, thermodynamic, and optics) and the dataset is diverse as well. This diverse and curated dataset is then used for training via direct preference optimization on the four level of granularity. The method is evaluated on physics-based benchmarks VideoPhy, PhyGenBench and VBench and shows significant improvements.

The core contributions of the work are: 1) compared to the previous work that explore only instance level alignment, this work explores a more fine-grained approach to ensure alignment, 2) The work presents an automated approach to curate a diverse dataset of video showcasing physical laws which removes the efforts of curating own data from scratch.

**Questions:**

Questions:
While the metrics used in the paper are inspired from the physics-based benchmarks, can the authors discuss how and why these metrics are good enough to evaluate the physics of the video? To my understanding, they are heavily based on human evaluation and motion accuracy, but I am unclear if they can efficiently evaluate the accuracy of the physics in the video.

Suggestions:
1. The metrics used in the paper can be explained in more detail. Some lines from the supplementary can be moved to the main paper to explain about the metrics used.

**Ethical Concerns:**

["NO or VERY MINOR ethics concerns only"]

**Final Justification:**

During the rebuttal phase, the authors clarified all my concerns. I would thus like to maintain my positive score.

**Limitations:**

Yes. Section F of the supplementary material addresses the ethical implications clearly.

**Quality:**

3

**Strengths And Weaknesses:**

Strengths:

1. The videos in the supplementary show improvement when compared with CogVideoX and show that the method is learning physical plausibility.
2. The contribution of presenting an automated method to curate a diverse video dataset having physical laws of dynamic objects, optics, and thermodynamics adds strength to the work.

Weakness:
The work uses four levels of granularity - instance, state, motion, and semantic. The semantic loss ensures that there is alignment between the text prompt and the video while the motion loss ensures the video's physical motion. For prompts such as 'a ceramic cup vanishes into thin air' in Figure 6, there can be a possibility that the model tries to over-optimize the semantic loss to ensure the alignment between the text prompt and video, finds a shortcut/cheat and doesn't actually optimize the motion loss much (cheats the motion loss objective). In my opinion, the motion loss is quite important to ensure the physical motion accuracy. I am curious to know how the proposed method ensures that the semantic loss doesn't become the primary/dominating loss function and finds a sub-optimal solution, and how the method ensures that the motion loss is also penalizing the network (given the loss weights are almost equal, 0.3 for motion and 0.2 for semantic).

---

> ### Author Rebuttal · Authors · 2025-07-31
>
> Thank you to all the reviewers for your thoughtful and constructive comments! We are truly encouraged to see that the reviewers appreciate several positive aspects of our work, such as **strong performance** (Reviewer `epiP`, `EaEb`, `JHEX`, `R9Wj`), **comprehensive & convincing experiments** (Reviewer `epiP`, `R9Wj`), **novel & effective framework** (Reviewer `R9Wj`, `EaEb`), **valuable & practical data selection pipeline** (Reviewer `R9Wj`, `EaEb`, `JHEX`), **well-motivated methodology** (Reviewer `epiP`, `JHEX`), and **clear writing** (Reviewer `epiP`).
>
> Thank you for your insightful feedback and expertise, which have guided us in improving the clarity and quality of our paper. Below, we provide detailed responses to your comments.
>
> ---
> > `Weakness`: How does the method ensure that the semantic loss doesn’t dominate and lead to sub-optimal solutions, while the motion loss effectively penalizes the network, given their similar weights (0.3 for motion and 0.2 for semantic)?
>
> Thank you for raising this important concern! We acknowledge that Motion Loss and Semantic Loss are optimized simultaneously during training, which could theoretically lead to the model overfitting to Semantic Loss and neglecting Motion Loss. Below, we provide a detailed explanation of how our framework mitigates this issue:
>
> - **Independent Loss Design:** Motion Loss and Semantic Loss operate on distinct inputs, where Motion Loss targets video structural information (*i.e.*, optical flow), while Semantic Loss focuses on text-video alignment. The physical characteristics captured by Motion Loss could not be "cheated" by optimizing Semantic Loss.
> - **Positive-Negative Sample Design:** Both losses rely on strict positive-negative sample comparisons. Even if Semantic Loss is over-optimized, Motion Loss penalizes discrepancies in physical motion between positive and negative samples, ensuring motion accuracy remains a priority.
> - **Experimental Validation:** Here we conduct an ablation on impossible prompts (IPV-TXT in `Figure 5`) to evaluate the impact of Motion Loss and Semantic Loss individually.
>
>     *Table A: Robustness Testing on IPV-TXT*
>     ||Visual Quality|Prompt Following|
>     |-|-|-|
>     |**PhysHPO (Ours)**|**58.82**|**47.09**|
>     |w/o $L_\text{Motion}$|54.06|44.93|
>     |w/o $L_\text{Semantic}$|55.39|43.75|
>     |Baseline|36.54|30.12|
>
>     Results show that removing either loss significantly reduces the model’s ability to generate impossible videos. Both are critical for handling complex prompts, demonstrating their complementary roles in the optimization process.
>
> Overall, the framework is designed to ensure that Motion Loss remains critical for physical motion accuracy while Semantic Loss complements it without dominating the optimization process. We appreciate your thoughtful feedback and hope this clarifies our approach!
>
> ---
> > `Question`: Can the authors discuss how and why these metrics are good enough to evaluate the physics of the video? To my understanding, they are heavily based on human evaluation and motion accuracy, but I am unclear if they can efficiently evaluate the accuracy of the physics in the video.
>
> Thank you for your thoughtful question! Our work utilizes **Semantic Adherence (SA)** and **Physical Commonsense (PC)** as core metrics to evaluate the physics of the generated videos, which are also widely adopted in recent works [1-3]. PC goes beyond simple motion accuracy by assessing adherence to real-world physical laws and intuitive commonsense, such as conservation of mass or gravity, while SA ensures alignment between the video content and the textual description. Together, these metrics provide a comprehensive framework for evaluating physical plausibility and semantic accuracy.
>
> Importantly, our evaluation uses VideoCon-Physics from VideoPhy for **automated** assessment, ensuring **scalability** and **objectivity**. However, we acknowledge that automated metrics have certain limitations, such as challenges in fully aligning with human preferences. To address this, benchmarks often **separate different physical phenomena for evaluation** to improve accuracy. Additionally, we emphasize the importance of **qualitative results** (`Figure 1`,`4`,`6`,`7`,`15-18`) and **user studies** (`Figure 5`) to complement automated metrics. To mitigate potential biases, we also perform quantitative tests across **multiple** benchmarks (*i.e.*, VideoPhy, PhyGenBench, VBench, IPV-TXT).
>
> We hope this clarifies how our metrics are designed and their role in evaluating the physics of generated videos!
>
> ---
> > `Suggestion`: The metrics used in the paper can be explained in more detail. Some lines from the supplementary can be moved to the main paper to explain about the metrics used.
>
> Thank you for the suggestion! We will move key details about SA and PC from the supplementary to the main paper for better clarity in our revision.
>
> We appreciate your insightful comments again and hope this rebuttal addresses your questions. Please feel free to reach out with further suggestions to improve clarity.
>
> ---
> [1] PhyT2V: LLM-Guided Iterative Self-Refinement for Physics-Grounded Text-to-Video Generation [CVPR’25]
>
> [2] Articulated Kinematics Distillation from Video Diffusion Models [CVPR’25]
>
> [3] VLIPP: Towards Physically Plausible Video Generation with Vision and Language Informed Physical Prior [ICCV’25]

---

> > ### Comment · Reviewer_EaEb · 2025-08-07
> > **Rebuttal Acknowledgement**
> >
> > Thank you authors for the detailed rebuttal. I have read the responses in the rebuttal carefully and they clarify my concerns well. My concerns were mainly that how does the model ensure that the semantic loss doesn't become the primary loss function and that the motion loss is penalizing the model too. The authors response about the independent design, the distinct inputs to the loss function, and the further experiment evaluations clarifies this concern well. My second concern about the evaluation metrics was also addressed well. The authors response that the Physical Commonsense metric not only evaluate the motion but also the physical laws clarified my concern.
> >
> > The authors are encouraged to incorporate all the reviewers' suggestions to improve their paper writing which would ensure more clarity.

---

> > > ### Author Response · Authors · 2025-08-08
> > > **Thank you immensely!**
> > >
> > > Dear Reviewer EaEb,
> > >
> > > Thank you for your thoughtful comments and strong support of our work. We deeply value your insights and feedback on PhysHPO, which have been instrumental in improving the quality of our research. We will further refine our writing to present our work better. It is our honor to address your concerns, and we sincerely thank you once again.
> > >
> > > Warm regards,
> > >
> > > The Authors

---

### Official Review · Reviewer_R9Wj · 2025-07-03

**Clarity:** 3
**Significance:** 3
**Originality:** 3
**Rating:** 4
**Confidence:** 4

**Summary:**

This paper introduces PhysHPO, a novel framework for improving the physical plausibility of generated videos. The core contribution is a hierarchical preference optimization method that aligns video generation at four levels: instance, state, motion, and semantic. This fine-grained approach moves beyond existing coarse-grained methods. The paper also proposes a practical and automated data selection pipeline to curate a high-quality, physically relevant dataset from existing large-scale video-text sources, thereby avoiding expensive manual dataset creation. Extensive experiments demonstrate that PhysHPO significantly enhances physical fidelity and overall quality on various benchmarks.

**Questions:**

1. In your ablation study (Table 3), "Only w/ L_instance" outperforms "Vanilla DPO w/ Our Data". Your instance-level loss uses a combination of err and gap negative samples. Does the "Vanilla DPO" baseline also use this dual-negative sampling strategy, or does it use a more standard approach of generating a single negative sample from the prompt? Clarifying this would help to precisely attribute the performance gains, distinguishing between the improvements from your specific instance-level implementation and the further improvements from the full hierarchical framework.

2. The "in-depth evolving prompting" strategy for physical fidelity scoring is a central part of your data selection pipeline. This process seems sensitive to the capabilities of the LLM used (Qwen2.5). Could you provide more insight into the robustness of this selection process?

3. The finding that PhysHPO generalizes better to "impossible prompts" is fascinating and suggests a deeper understanding beyond rote memorization of physics. You hypothesize that enhancing physical fidelity equips the model with a stronger ability to generalize. Could you elaborate on this?

**Ethical Concerns:**

["NO or VERY MINOR ethics concerns only"]

**Final Justification:**

After reviewing the author's rebuttal, almost all my concerns have been resolved. Therefore, I would like to keep my original positive score.

**Limitations:**

Yes

**Quality:**

3

**Strengths And Weaknesses:**

**Strengths:**

1. The proposed PhysHPO framework is a significant and intuitive contribution to video generation. By decomposing the abstract goal of "physical plausibility" into four concrete, hierarchical levels of alignment (Instance, State, Motion, Semantic) , the paper presents a much more sophisticated and fine-grained approach than prior work on Direct Preference Optimization (DPO) in video, which has largely focused on coarse, instance-level alignment. This hierarchical structure is well-justified and provides a clear path to improving specific aspects of video realism.

2. The proposed automated data selection pipeline is a clever and highly valuable contribution. The multi-stage process—filtering for real-world videos , scoring for physical fidelity using an "in-depth evolving prompting" strategy with LLMs , and ensuring diversity —is both novel for video generation  and shown to be highly effective

3. The authors have conducted an extensive and convincing set of experiments. They apply their method to multiple advanced, large-scale video models (CogVideoX-2B/5B, Hunyuan Video) and evaluate on a diverse suite of benchmarks, including both physics-focused (VideoPhy, PhyGenBench) and general quality (VBench)  benchmarks. They consistently demonstrate superior performance over strong baselines, including vanilla DPO and test-time optimization methods like PhyT2V.

**Weaknesses:**

1. While the performance improvements shown in the tables are often substantial, the stochastic nature of video generation and automated evaluation metrics means that the robustness of these gains is not statistically validated.

2.  The paper does a good job of trying to isolate variables, for instance by applying vanilla DPO on the same selected data. The ablation in Table 3 shows that "Only w/ L_instance" (the authors' instance-level method) outperforms "Vanilla DPO". This suggests the instance-level contribution alone provides a boost over the baseline DPO. It would strengthen the paper to more clearly disentangle the gains from the novel instance-level formulation and the gains from the additional hierarchical levels (State, Motion, Semantic).

3. While the paper demonstrates the effectiveness of this approach, it also introduces a dependency on these models. The cost, reproducibility, and potential biases of this "LLM-as-a-Judge" paradigm  are not deeply discussed. The selection process is defined by a fixed set of 17 physical phenomena from a prior work, which may limit the breadth of physical concepts the pipeline can identify. A discussion on the sensitivity to the choice of LLM or the physical taxonomy would be beneficial.

---

> ### Author Rebuttal · Authors · 2025-07-31
>
> Thank you to all the reviewers for your thoughtful and constructive comments! We are truly encouraged to see that the reviewers appreciate several positive aspects of our work, such as **strong performance** (Reviewer `epiP`, `EaEb`, `JHEX`, `R9Wj`), **comprehensive & convincing experiments** (Reviewer `epiP`, `R9Wj`), **novel & effective framework** (Reviewer `R9Wj`, `EaEb`), **valuable & practical data selection pipeline** (Reviewer `R9Wj`, `EaEb`, `JHEX`), **well-motivated methodology** (Reviewer `epiP`, `JHEX`), and **clear writing** (Reviewer `epiP`).
>
> We greatly appreciate your detailed review and valuable insights, which help us refine our manuscript. Here, we respond to your comments systematically to address your concerns.
>
> ---
> > `Weakness 1`: While the performance improvements shown in the tables are often substantial, the stochastic nature of video generation and automated evaluation metrics means that the robustness of these gains is not statistically validated.
>
> Thank you for your thoughtful question! You are correct that the stochastic nature of video generation and automated evaluation metrics may not fully capture human preferences. This is precisely why presenting sufficient **qualitative results** (`Figure 1`,`4`,`6`,`7`,`15-18`) and conducting **User Studies** (`Figure 5`) are critical to complement quantitative analyses. Additionally, we have evaluated our framework across **multiple** benchmarks (*i.e.*, VideoPhy, PhyGenBench, VBench, IPV-TXT) to minimize potential biases introduced by these limitations.
>
> ---
> > `Weakness 2`: It would strengthen the paper to more clearly disentangle the gains from the novel instance-level formulation and the gains from the additional hierarchical levels (State, Motion, Semantic).
>
> Thanks for your thoughtful suggestion! `Table 3` progressively removes each Level loss to show their contributions. While our initial version didn’t compare individual Level alignments to vanilla DPO due to time and computational costs, we now provide this ablation:
>
> *Table A: Additional Ablation Study on Level Losses on VideoPhy*
> ||SS|SF|FF|Overall|
> |-|-|-|-|-|
> |Vanilla DPO w/ Our Data|28.2|50.0|51.8|41.3|
> |+ $L_\text{Instance}$|28.9|50.7|51.8|41.9|
> |+ $L_\text{State}$|29.6|50.0|53.6|42.2|
> |+ $L_\text{Motion}$|30.3|50.7|55.4|43.0|
> |+ $L_\text{Semantic}$|28.9|50.0|53.6|41.9|
>
> ---
> > `Weakness 3`: (1) The cost, reproducibility, and potential biases of this "LLM-as-a-Judge" paradigm are not deeply discussed. (2) The selection process is defined by a fixed set of 17 physical phenomena from a prior work, which may limit the breadth of physical concepts the pipeline can identify. A discussion on the sensitivity to the choice of LLM or the physical taxonomy would be beneficial.
>
> (1) **Dependency on the "LLM-as-a-Judge" Paradigm:** Using LLMs significantly improves **efficiency** compared to human annotators, as they can rapidly process video captions and score across 17 dimensions. To address potential biases, we employ an in-depth evolving prompting strategy (`Figure 8`), which guides the LLM through **step-by-step reasoning** and requires it to output both scores and justifications. This reduces hallucinations and ensures **effective** scoring. Furthermore, as shown in `Figures 10` and `11`, our pipeline achieves **over 90% consistency** with human annotations while being cost-effective.
>
> (2) **Scope of the 17 Physical Phenomena:** While LLMs can analyze more granular phenomena, the 17 selected phenomena comprehensively represent core **Dynamic**, **Thermodynamic**, and **Optic** categories. This taxonomy balances breadth and clarity, ensuring interpretability for both LLMs and human annotators without introducing **excessive complexity**. Here, we show a comparison between our 17 phenomena and the categories generated by the LLM itself:
>
> *Table B: Adjacency Accuracy ($\pm$ 0.3) of Different Scoring Strategies with Human Scoring (Following `Figure 11`'s Setting)*
> ||Dynamic|Thermodynamic|Optic|
> |-|-|-|-|
> |LLM Self-Play|72.0|76.0|81.0|
> |Direct Scoring on 3 Categories|74.0|69.0|76.0|
> |**17 Phenomena (Ours)**|**92.0**|**90.0**|**91.0**|
>
>
> The results demonstrate that our taxonomy effectively captures the most critical aspects while avoiding overly fragmented or ambiguous classifications. We hope this addresses your concerns and welcome further feedback.
>
> ---
> > `Question 1`: Does the "Vanilla DPO" baseline also use this dual-negative sampling strategy, or does it use a more standard approach of generating a single negative sample from the prompt?
>
> Thank you for your detailed review and thoughtful question! **"Vanilla DPO w/ Our Data"** uses the standard **one-positive *vs.* one-negative** paradigm, where the positive video is $y_w$ and the negative video is $y_l^{err}$ in our data. The comparison with **"Only w/ $L_\text{instance}$"** highlights the additional contribution of modeling the semantic gap ($y_l^{gap}$) to performance improvements.
>
> ---
> > `Question 2`: The "in-depth evolving prompting" strategy for physical fidelity scoring is a central part of your data selection pipeline. This process seems sensitive to the capabilities of the LLM used (Qwen2.5). Could you provide more insight into the robustness of this selection process?
>
> Thank you for your insightful question! The capabilities of the base model indeed play a significant role in the robustness of the physical fidelity scoring process. As shown in **Table C**, we compare the impact of using different LLMs on PhysHPO’s performance, highlighting the variability introduced by different base models.
>
> *Table C: Performance Comparison between Different LLMs on VideoPhy*
> ||SS|SF|FF|Overall|
> |-|-|-|-|-|
> |w/ LLaMA3-8B|27.5|51.4|48.2|41.0|
> |w/ DeepSeek-V3|29.6|**54.1**|**60.7**|45.1|
> |**w/ Qwen2.5 (Ours)**|**32.4**|**54.1**|58.9|**45.9**|
>
> Furthermore, **Table D** demonstrates how both Qwen2.5 and DeepSeek-V3, combined with our in-depth evolving prompting strategy, achieves better alignment with human evaluations. This strategy leverages specific contextual comparisons to guide the model towards more professional and detailed assessments of physical fidelity.
>
> *Table D: Human Align Accuracy with Qwen 2.5 and DeepSeek-V3*
> ||Dynamic|Thermodynamic|Optic|
> |-|-|-|-|
> |Qwen2.5|74.0|69.0|76.0|
> |**+ In-Depth Evolving Prompting (Ours)**|**92.0**|**90.0**|**91.0**|
> |DeepSeek-V3|69.0|73.0|71.0|
> |**+ In-Depth Evolving Prompting (Ours)**|**90.0**|**91.0**|**89.0**|
>
> ---
> > `Question 3`: The finding that PhysHPO generalizes better to "impossible prompts" is fascinating and suggests a deeper understanding beyond rote memorization of physics. You hypothesize that enhancing physical fidelity equips the model with a stronger ability to generalize. Could you elaborate on this?
>
> Thank you for your thoughtful question! Our experiments show that PhysHPO generalizes better to "impossible prompts". We believe this may stem from **physical fidelity going beyond simple text-pixel alignment**, enabling the model to learn deeper, more abstract relationships between physical and semantic contexts.
>
> By modeling the boundaries of physics, PhysHPO can creatively explore these limits while maintaining semantic coherence. Its hierarchical alignment strategy further integrates physical understanding with context, allowing logical and consistent generation even in implausible scenarios.
>
> This suggests that enhancing physical fidelity optimizes the model’s knowledge structure, strengthening its ability to reason and generalize beyond rote memorization.
>
> We appreciate your interest in this aspect of our work!
>
> ---
> Thank you again for your valuable feedback. We hope this rebuttal resolves your concerns and enhances the clarity of our work. We remain open to any further suggestions for improvement.

---

### Official Review · Reviewer_epiP · 2025-07-03

**Clarity:** 3
**Significance:** 3
**Originality:** 3
**Rating:** 4
**Confidence:** 4

**Summary:**

The paper introduces a hierarchical DPO framework that enhances video generation physical plausibility through multi-granularity preference alignment (instance, boundary, motion, semantic). The work also proposes an automated data selection pipeline to select real-world, physically informative, and diverse video-text data from existing large-scale datasets, instead of relying on resource-intensive dataset construction.

**Questions:**

- Although improvements are shown, the main text lacks systematic analyses of failure cases. where does PhysHPO still fail to produce physically plausible videos, or what are the remaining challenges?
- Does the filtering pipeline's effectiveness depend heavily on threshold choices and LLM/VLM selection, and if so, how significantly do these parameters impact final model performance?
- Does the framework maintain its effectiveness for non-physical domains (e.g., stylized content), and are these edge cases systematically evaluated in the experiments?

**Ethical Concerns:**

["NO or VERY MINOR ethics concerns only"]

**Limitations:**

Yes

**Quality:**

3

**Strengths And Weaknesses:**

Strengths:

- The writing is mostly clear and easy to follow.
- The four-level hierarchical approach (instance, state, motion, semantic) for DPO is well-motivated and clearly described, addressing core limitations of prior work that only considered coarse alignment.
- The paper provides comprehensive experimental validation on multiple meaningful, recent benchmarks (VideoPhy, PhyGenBench, VBench) , demonstrating better performance compared to relevant baselines approaches.

Weaknesses:

- Hierarchical modeling may increase training complexity or cost. There is no clear discussion in the main text regarding the computational demands for the hierarchical DPO approach relative to vanilla DPO or SFT approaches.
- The approach employs LLM/VLM-based scoring and captioning for data selection and semantic preference, but does not thoroughly assess how model choice impacts reliability, e.g., whether different foundation models would yield varying downstream performance.

---

> ### Author Rebuttal · Authors · 2025-07-31
>
> Thank you to all the reviewers for your thoughtful and constructive comments! We are truly encouraged to see that the reviewers appreciate several positive aspects of our work, such as **strong performance** (Reviewer `epiP`, `EaEb`, `JHEX`, `R9Wj`), **comprehensive & convincing experiments** (Reviewer `epiP`, `R9Wj`), **novel & effective framework** (Reviewer `R9Wj`, `EaEb`), **valuable & practical data selection pipeline** (Reviewer `R9Wj`, `EaEb`, `JHEX`), **well-motivated methodology** (Reviewer `epiP`, `JHEX`), and **clear writing** (Reviewer `epiP`).
>
> Your expertise and thoughtful review have been instrumental in improving our manuscript. Below, we address your comments point-by-point to clarify and strengthen our work.
>
> ---
> > `Weakness 1`: Hierarchical modeling may increase training complexity or cost. There is no clear discussion in the main text regarding the computational demands for the hierarchical DPO approach relative to vanilla DPO or SFT approaches.
>
> Thank you for pointing out this important aspect. We acknowledge that hierarchical modeling increases computational complexity, but it also brings significant performance gains. Balancing this trade-off presents a challenge and an opportunity for developing more efficient, scalable approaches, which we will also further discuss in future versions.
>
> ---
> > `Weakness 2`: The approach employs LLM/VLM-based scoring and captioning for data selection and semantic preference, but does not thoroughly assess how model choice impacts reliability.
>
> Thank you for highlighting this issue! Different foundation models indeed impact downstream performance. Here we compare several VLM in our pipeline:
>
> *Table A: Performance Comparison between Different VLMs on VideoPhy*
> ||SS|SF|FF|Overall|
> |-|-|-|-|-|
> |w/ LLaMA3-8B|27.5|51.4|48.2|41.0|
> |w/ DeepSeek-V3|29.6|**54.1**|**60.7**|45.1|
> |**w/ Qwen2.5 (Ours)**|**32.4**|**54.1**|58.9|**45.9**|
>
> Qwen2.5, a widely used open-source model, outperforms others, emphasizing the importance of choosing suitable LLMs and prompts for task-specific pipelines. Variations in architecture and pretraining data explain these differences, with models like Qwen2.5 excelling in multimodal reasoning tasks.
>
>
> ---
> > `Question 1`: Although improvements are shown, the main text lacks systematic analyses of failure cases. where does PhysHPO still fail to produce physically plausible videos, or what are the remaining challenges?
>
> Thank you for pointing this out. PhysHPO, as a strategy to enhance the physical plausibility of VDM-generated videos, has demonstrated strong performance and deeper learning from real-world videos. However, a systematic analysis of failure cases reveals certain limitations. Specifically, **high-speed human action videos**, such as **dancing** or **gymnastics**, often fail to capture precise motion dynamics and physical plausibility. This is also largely due to limitations in the **base model**’s ability to represent complex, rapid movements, as well as potential gaps in the **quality** and **quantity** of the training data used.
>
> These challenges highlight areas for improvement and remain open questions for further exploration by the research community. We recognize the importance of thoroughly analyzing failure cases and will include a more **detailed discussion** in a future version of this work.
>
> ---
> > `Question 2`: Does the filtering pipeline's effectiveness depend heavily on threshold choices, and if so, how significantly do these parameters impact final model performance?
>
> Yes, the effectiveness of the filtering pipeline does depends on threshold choices, which could impact the quality and relevance of selected data. While `Figure 3` provides a preliminary comparison, here we present a more detailed analysis:
>
> *Table B: Diversity Selection Threshold Analysis on PhyGenBench*
> ||Mech.|Opt.|Ther.|Mate.|Overall|
> |-|-|-|-|-|-|
> |0.0|0.50|0.66|0.43|0.60|0.56|
> |0.3|0.53|0.66|0.47|0.60|0.58|
> |0.6|0.53|0.68|0.50|0.63|0.59|
> |**0.9 (Ours)**|**0.55**|**0.68**|**0.50**|**0.65**|**0.61**|
>
> It clearly demonstrates how selecting appropriate thresholds is crucial for ensuring high-quality data selection and improving downstream task performance.
>
> ---
> > `Question 3`: Does the framework maintain its effectiveness for non-physical domains (e.g., stylized content), and are these edge cases systematically evaluated in the experiments?
>
> Thank you for your thoughtful question! We have considered this important evaluation criterion and used VBench as the benchmark for non-physical domains, including 16 metrics such as **Appearance Style** (AS), as shown in below:
>
> *Table C: Full Dimensions of VBench Evaluation*
> ||Total Score|Quality Score|Semantic Score|SS|BC|TF|MS|DD|AQ|IQ|OC|MO|HA|Color|SR|Scene|AS|TS|OC|
> |-|-|-|-|-|-|-|-|-|-|-|-|-|-|-|-|-|-|-|-|
> |CogVideoX|81.9|83.1|77.3|96.5|96.7|99.0|97.2|69.5|61.9|63.3|85.1|63.9|98.6|83.0|68.9|52.0|25.0|25.4|27.7|
> |+PhyT2V|82.3|83.3|78.3|96.7|**97.0**|**99.0**|97.3|69.6|62.0|63.5|86.9|64.9|99.2|83.7|71.0|52.3|24.8|26.3|27.7|
> |+Vanilla DPO|82.4|83.3|78.7|96.8|96.8|98.4|97.3|70.2|63.3|63.8|87.2|65.1|99.2|83.9|72.0|53.1|24.9|26.3|27.9|
> |**+PhysHPO (Ours)**|**82.8**|**83.7**|**79.3**|**96.8**|96.9|98.6|**97.5**|**70.3**|**63.5**|**64.8**|**87.4**|**65.4**|**99.5**|**83.9**|**72.2**|**54.1**|**25.7**|**26.5**|**27.9**|
>
> The results demonstrate that PhysHPO not only significantly enhances the physical plausibility of videos generated by the base model but also maintains or even improves its performance in non-physical (general) domains.
>
> ---
> We hope this rebuttal sufficiently addresses your concerns and clarifies the points raised. We are open to further discussion and greatly appreciate your thoughtful feedback.

---

> ### Author Response · Authors · 2025-08-08
> **A Gentle Reminder of Feedback**
>
> Dear Reviewer epiP,
>
> This is a gentle reminder that **the discussion phase will end in less than 2 days**, and we have not yet received your feedback on our rebuttal. We fully understand the demands on your time and sincerely appreciate your efforts in reviewing our work. To facilitate communication, we’ve summarized our key responses below:
>
> - **Computational Complexity (`Weakness 1`):** We acknowledged the increased training complexity of hierarchical modeling, emphasizing the trade-off between computational demands and significant performance gains. We will explore optimization methods in future work.
> - **Model Reliability (`Weakness 2`):** We compared several LLMs in our pipeline, demonstrating that Qwen2.5 excels in the reasoning tasks.
> - **Failure Cases (`Question 1`):** We systematically analyzed failure cases, identifying challenges in high-speed human action videos (*e.g.*, dancing, gymnastics).
> - **Threshold Sensitivity (`Question 2`):** We analyzed the impact of threshold choices on the filtering pipeline’s effectiveness.
> - **Effectiveness in Non-Physical Domains (`Question 3`):** Using VBench, we demonstrated that PhysHPO maintains or improves performance in non-physical domains, ensuring robustness across diverse content types.
>
> Thank you once again for your thoughtful insights and dedication throughout the review process. Your insights deeply inspired us, and **we are genuinely committed to addressing your concerns with the utmost care**. We would be greatly grateful if you could let us know **whether our rebuttal has sufficiently addressed your concerns**.
>
> Thanks and Regards,
>
> The Authors

---

### Author Response · Authors · 2025-08-07
**Rebuttal Progress Summary & Sincere Invitation to Discussion**

**Dear Reviewers,**

Warm greetings to you! We deeply appreciate your thorough and constructive feedback, which has been invaluable in improving our manuscript. Below, we summarize the key concerns raised and our corresponding responses:

1. **Model Choice Sensitivity in Data Selection (`Reviewers epiP, R9Wj`):** We analyzed the impact and robustness of LLMs within data selection, highlighting Qwen2.5’s advantages.
2. **Failure Cases and Generalization (`Reviewers epiP, R9Wj`):** We examined failure cases in high-speed action videos and elaborated on PhysHPO’s generalization to "impossible prompts".
3. **Automated Evaluation Metrics (`Reviewers R9Wj, EaEb`):** We further explained the rationale behind automated metrics and underscored the importance of qualitative results and user studies.
4. **Hierarchical Levels and Loss Balance (`Reviewers R9Wj, EaEb, JHEX`):** We conducted additional ablation studies to isolate the contributions of hierarchical levels and clarified strategies for balancing loss functions.
5. **Vanilla DPO w/ Our Data (`Reviewers R9Wj, JHEX`):** We clarified the adaptation process of positive/negative samples in vanilla DPO with our data, addressing their role in comparative analyses.
6. **Writing and Equation Clarity (`Reviewer JHEX`):** We committed to revising writing and explicitly detailing equations for better clarity and alignment with DPO.

Thank you once again for your valuable feedback. **With the utmost humility, we kindly extend an invitation for further discussion** before the deadline. We would be honored and grateful to hear your thoughts.

Sincerely,

The Authors

---

### Note · Authors · 2025-08-12

Dear Program Committee,

Warm greetings to you in mid-August! We sincerely thank all reviewers, the Area Chair, the Senior Area Chair, and Program Chairs for your time, effort, and constructive feedback.

We are truly encouraged by the reviewers’ recognition of several positive aspects of our work, including its **strong performance** (Reviewers `epiP`, `EaEb`, `JHEX`, `R9Wj`), **comprehensive & convincing experiments** (Reviewers `epiP`, `R9Wj`), **novel & effective framework** (Reviewers `R9Wj`, `EaEb`), **valuable & practical data selection pipeline** (Reviewers `R9Wj`, `EaEb`, `JHEX`), **well-motivated methodology** (Reviewers `epiP`, `JHEX`), and **clear writing** (Reviewer `epiP`). This positive feedback has greatly motivated us and reaffirmed the significance of our contributions.

During the rebuttal and discussion phases, we have worked diligently to address concerns raised during the review process, which has significantly enhanced our paper, including **model choice sensitivity** (`epiP`, `R9Wj`), **failure cases & generalization** (`epiP`, `R9Wj`), **automated evaluation metrics** (`R9Wj`, `EaEb`), **hierarchical levels & loss balance** (`R9Wj`, `EaEb`, `JHEX`), **vanilla DPO adaptation** (`R9Wj`, `JHEX`), and **writing clarity** (`JHEX`).

We are particularly grateful to Reviewers `epiP`, `R9Wj`, and `EaEb` for their positive evaluations and to Reviewers `R9Wj` and `EaEb` for their active engagement during the discussions. We are **honored that our rebuttal successfully addressed their concerns**. However, despite our best efforts, we regret that we did not receive confirmation from Reviewers `epiP` and `JHEX` on whether their concerns were fully resolved.

For Reviewer `JHEX`, whose feedback highlighted areas for improvement in the initial phase, we regret that certain writing details in our manuscript may have caused confusion. We have provided further clarifications and are committed to refining these aspects to improve clarity and alignment, and we would welcome any opportunity for further discussion to address remaining concerns.

Finally, we sincerely thank all reviewers for their thoughtful evaluations, and we are deeply appreciative of the Area Chair’s guidance and support throughout this process. Thank you!

Warm regards,

The Authors

---

### Decision · Program_Chairs · 2025-09-17

**Decision:**

Accept (poster)

**Comment:**

This paper introduces PhysHPO, a hierarchical direct preference optimization framework that improves physical plausibility in video generation.

All four reviewers gave positive ratings, recognizing the paper's strong performance, comprehensive experiments, novel hierarchical framework, and valuable automated data selection pipeline. The reviewers particularly appreciated the well-motivated methodology that decomposes physical plausibility into four hierarchical levels (instance, state, motion, semantic) and the practical contribution of an automated data curation process that eliminates the need for expensive manual dataset construction.

The main concerns raised during the initial review phase centered on computational complexity, model choice sensitivity in the data selection pipeline, failure case analysis, the balance between different loss components, and clarity of mathematical formulations. The authors provided a thorough rebuttal addressing each of these points with additional ablation studies, comparisons across different language models, detailed explanations of loss design, and clarifications on implementation details.

Following the rebuttal, reviewers R9Wj and EaEb explicitly acknowledged that their concerns had been resolved and confirmed their positive ratings. While reviewers epiP and JHEX did not provide final confirmations, reviewer JHEX's final justification stated that "the authors have addressed most of my concerns" and recommended working on writing clarity, indicating general satisfaction with the responses.

The experimental evaluation is comprehensive, covering multiple advanced video models and diverse benchmarks including both physics-focused (VideoPhy, PhyGenBench) and general quality (VBench) evaluations. The consistent improvements over strong baselines, including vanilla DPO and specialized methods like PhyT2V, demonstrate the effectiveness of the approach.